# Dataset of Georeferenced Dams in South America (DDSA)

Bolivar Paredes-Beltran[1-2], Alvaro Sordo-Ward[1], Luis Garrote[1]

[1] Department of Civil Engineering: Hydraulics, Energy and Environment, Universidad Politécnica de Madrid, Madrid, 28040, Spain

[2] Carrera de Ingeniería Civil, Facultad de Ingeniería Civil y Mecánica, Universidad Técnica de Ambato, Ambato, 180206, Ecuador

*Correspondence to*: be.paredes@alumnos.upm.es, be.paredes@uta.edu.ec

**Abstract.** Dams and their reservoirs generate major impacts on society and the environment. In general, its relevance relies on facilitating the management of water resources for anthropogenic purposes. However, dams could also generate many

potential adverse impacts related to safety, ecology or biodiversity. These factors, and the additional effects that climate change could cause in these infrastructures and their surrounding environment, highlight the importance of dams and the necessity for their continuous monitoring and study. There are several studies examining dams both at regional and global scale, however, those that include the South America region focus mainly on the most renowned basins (primarily the Amazon basin), most likely due to the lack of records on the rest of the basins of the region. For this reason, a consistent database of georeferenced

dams located in South America is presented: Dataset of georeferenced dams in South America DDSA. It contains 1,010 entries of dams with a combined reservoir volume of 1,017 cubic kilometres and it is presented in form of a list describing a total of 24 attributes that include the dams name, characteristics, purposes and georeferenced location. Also, hydrological information on the dams' catchments is also included: catchment area, mean precipitation, mean near-surface temperature, mean potential evapotranspiration, mean runoff, catchment population, catchment equipped area for irrigation, aridity index, residence time

and degree of regulation. Information was obtained from public records, governments records, existing international databases and from extensive internet research. Each register was validated individually and geolocated using public access online map browsers and then, hydrological and additional information was derived from a hydrological model computed using the HydroSHEDS dataset. With this database, we expect to contribute to the development of new research in this region. The database is publicly available in https://doi.org/10.5281/zenodo.4315647 (Paredes-Beltran et al., 2020).

## 1 Introduction

Dams and their reservoirs provide continuous water supply for different anthropogenic necessities such as electricity generation, water supply, irrigation, flood control, livestock feed or recreation. This becomes crucial in areas where water resources are scarce either by seasonality or due to the increasing effects of climate change. However, in many cases dams and their reservoirs are controversial because they can cause acute and chronic impacts in the environment and also in the nearby human

settlements. These impacts are generally well known and include the modification of aquatic and terrestrial ecosystems,

reduction of biodiversity, changes in the morphology of river systems, degradation of water quality and characteristics, alterations in sediments and nutrients discharge, changes in seasonal hydrological regimes, the migration of human settlements or changes in land-use patterns (Barbarossa et al., 2020; Bednarek, 2001; Nilsson et al., 2005; Pekel et al., 2016; Stoate et al., 2009).

Due to the obvious importance of dams and their reservoirs, continuous monitoring and resources needs to be dedicated on these structures. The importance of dams and reservoirs also makes them relevant for research. For example, there are studies that assess or propose improvements on construction methods for dams (Ladd, 1992; Noorzaei et al., 2006; Xu et al., 2012), examine improvements on monitoring the structural health or safety of the dam (Gabriel-Martin et al., 2017; Li et al., 2004; Sjödahl et al., 2008) or evaluate their behaviour during seismic or failure events (Alonso et al., 2005; Zabala and Alonso,

2011). Reservoirs associated with dams are also relevant, for instance, by examining the effects, impacts and management alternatives of sediments fluxes (Dai and Liu, 2013; Kondolf et al., 2014). Usually, these studies require knowing a minimum set of characteristics of the dam, including their location and in most of the cases, need to be included into hydrological models. The influence and effects of dams on their surrounding environments is also relevant for research. For instance, the impacts caused by dams and reservoirs on water supply (Biemans et al., 2011; Bouwer, 2000; Khalkheili and Zamani, 2009), the

potential effects of climate change on altered river networks (Döll et al., 2009; Nilsson et al., 2005), the prospective scenarios that climate change could cause on irrigation water (Chavez-Jimenez et al., 2015; Elliott et al., 2014; Garrote et al., 2015), the repercussions of dams on water resources and biodiversity (Bejarano et al., 2017; Liermann et al., 2012; Vörösmarty et al., 2010) or the hydrological alterations caused by dams and reservoirs (Batalla and Go, 2004; Ibàñez and Prat, 1996).

In South America, relevant studies about dams at a full regional scale are rather scarce and usually focus on aquatic biodiversity

conservation (Barletta et al., 2010; Reis et al., 2016) or river segmentation (Castello et al., 2013; Fearnside, 2001; Latrubesse et al., 2017; Roberto et al., 2009) and in most of the cases their conclusions highlight that potential negative effects of dams are low to moderate. However, these studies generally present two important limitations when trying to reach a full region scale: first, these focus only on the most relevant or renowned basins such as the Amazon, Paraná - La Plata or Orinoco, and second, these only consider a limited amount of dam records.

There are several published dam databases that include information from South America. The largest and most recognized database is the World Register of Dams published by the International Commission on Large Dams (ICOLD, 2020) which reports 1,922 dams entries for South America; nonetheless, this database is not georeferenced which limits its use. AQUASTAT database was presented by (FAO, 2015) but it has not been updated since 2015 and for South America it only reports 344 entries of georeferenced dams. Finally, another relevant database is the GRaND database (Lehner et al., 2011) which has

been updated for the year 2019 and accounts for 343 geolocated dam entries for South America.

Here, we present an extensive and revised database with 1,010 registers of dams in South America, including information on their identification, the dam main characteristics, the dam purposes and their spatial location. Also, it includes hydrological information derived from the HydroSHEDS dataset (Lehner et al., 2008): catchment area, mean near-surface temperature, mean precipitation and mean potential evapotranspiration from the Climatic Research Unit (CRU) time-series dataset (Harris

et al., 2020), mean runoff from the University of New Hampshire Global Runoff Data Centre (GRDC) composite runoff field (Fekete et al., 2002), catchment population data from the Global Rural-urban Mapping Project (GRUMP) (Center for International Earth Science Information Network CIESIN et al., 2011), catchment equipped area for irrigation from the Global Map of Irrigated Area dataset (Siebert et al., 2005), aridity index, residence time and degree of regulation. This database has been developed to provide researchers additional information on dams, reservoirs and dams' catchments in South America,

with the expectation to further promote research on dams, hydrology, water resources, ecology environmental science, geography or sociology either on a local, regional or global scale.

This database is publicly available free for use in https://doi.org/10.5281/zenodo.4315647 (Paredes-Beltran et al., 2020).

## 2 Data description

### 2.1 Study Area

The study area is the continent of South America and includes Argentina, Bolivia, Brazil, Chile, Colombia, Ecuador, Guyana, French Guiana, Paraguay, Peru, Suriname, Uruguay and Venezuela. A total of 1,010 catchments were considered which drain an area of approximately 5,283,000 km$^2$ and discharge their waters to both the Pacific Ocean and the Atlantic Ocean. Within each of this catchments, necessary observations were made to accurately locate dams with their respective reservoirs.

The study area is diverse and full of contrasts due to its unique geography; for example, the Andes mountains are a continuously

seismic region that covers the entire west coast of the continent, the Amazon rainforest in the central part of the continent, large semiarid plains in the southeast and also the Atacama desert, which is a region of extreme aridity in the southwest. In the Andes we have the presence of large glaciers that mostly drain east to form several rivers, including some of the largest in the world such as the Amazon, the Paraná - Rio de la Plata and the Orinoco river. On the east coast of the continent, there exist humid mountain formations that extend from Venezuela to northern Brazil.

The climate on the continent is diverse mainly due to its size and topography, but also due to its wind patterns and ocean currents. Around the equator, climate can be considered mainly as tropical and humid with large amounts of rain, which decreases while moving further north and south of the equator, where different weather patterns are found. In the southern part of the continent, the humid winds of the Pacific Ocean provide rain to several areas in the coast of Chile, which are blocked due to the Andes mountains and causes low precipitation around the year in the Patagonia region in the southeast. The climate

within the Andes mountains is characterized as dry and cold and covers the highest mountains with snow all year round. The driest region in the continent is the Atacama Desert due to its almost zero humidity, and it is located in the north of Chile and the south of Peru.

Climate diversity in South America is also due to the occurrence of several interannual and interdecadal large-scale climate events. For example, the "El Niño Southern Oscillation" (ENSO) which is a Pacific Ocean sea-surface temperature (SST)

event that fluctuates from warm ("El Niño") and cold ("La Niña") phases, and occurs in periods of between two to seven years.

The ENSO causes disruptions of precipitation and temperature in the continent and is often considered as the major source of interannual climate variability in most of South America.

In general, the "El Niño" causes low precipitation over tropical South America, high precipitation over the south east of the region and high temperatures over tropical and subtropical areas. Also, the "El Niño" is often associated to regionally diverse events like droughts in the Amazon rainforest and the north-east of South America, but also to flooding events in the tropical west coast and the south-east of the continent (Cai et al., 2020; Hao et al., 2020). On the other hand, "La Niña" generally causes the opposite precipitation and temperature events for the same areas (Garreaud et al., 2009).

Other regional climate events in South America like the sea-surface temperature (SST) anomalies in the tropical Atlantic (Garreaud et al., 2009; Jiménez-Muñoz et al., 2016), the Pacific Decadal Oscillation (PDO) (Nathan and Steven, 2002), or the Antarctic Oscillation (AAO) and the North Atlantic Oscillation (NAO) (Garreaud et al., 2009) also play an important role in the variability of South America climate.

## 2.2 Data sources and assessments methods

### 2.2.1 Compilation of preliminary information

A preliminary compilation of data regarding dams and reservoirs in the continent was first carried out to serve as a basis prior to the creation of this database. For this, two types of bibliographic sources were used: first, dams and reservoirs information from currently published databases, and second, records available about dams, reservoirs and water resources, from governments and other official sources. In the first case, we used two well-known open access databases of dams and reservoirs: the GRaND database (http://globaldamwatch.org/grand/, last access: 23 May 2020) and the AQUASTAT database (http://www.fao.org/aquastat/es/databases/dams/, last access: 23 May 2020). In the second case, we found that many governments keep up-to-date and comprehensive records of their water resources including dams and reservoirs. However, there were cases in which official information is not available. Table 1 details the public sources from which most of the information was obtained for each of the countries.

After an extensive review, we determined that georeferenced information about dams in this continent is limited. This is one of the main reasons why we aimed to develop a new database that includes all the current consistent information available. We proceeded in three stages: first, we collected all the available published information on dams and reservoirs; second, we compared and validated this data with the existing information available from local and national governments; and finally, we determined the geolocation of each point. This information has been processed and we carried out an extensive data validation and error checking, elimination of duplicate or inaccurate entries and completion of information where possible.

First, we researched for the most relevant databases of dams and reservoirs available and found three consistent results: The World Register of Dams from ICOLD, the GRaND database and the AQUASTAT database of dams. After the initial inspection, we discarded the ICOLD database because even though it is widely considered as the largest database on dams with over 57,985 entries worldwide and 1,922 dam entries in South America, it is not georeferenced nor it is an open-access database,

which limits later validation of our results. Then, we inspected the AQUASTAT database (which has not been updated since 2015) and collected detailed information of more than 14,000 dams; nonetheless, in the case of South America the list consists

of 1,964 dams of which only 344 entries are georeferenced. Finally, we examined the GRaND database which presents 7,320 entries, however, only 343 of those entries correspond to South America.

Once initial information was collected from open-access databases to assemble our preliminary list, we examined public records available from local and national governments in each country. We compiled them in order to compare this data with our preliminary list, data collected from governments and other public sources is available in different formats and in most cases

required different types of approximation and treatment to obtain results. Each dam record was compared individually and in the case of correspondence it was accepted, in the case of countries where we did not find available public reports, we compared and verified our preliminary records with information available on the internet, focusing on dams with reservoir capacity greater than one cubic hectometre, although some records with smaller reservoir volume were included as these could be verified in a reliable manner.

Finally, a supplementary search on the internet was performed to exclude gaps, mismatches or errors.

### 2.2.2 Geolocation of entries

Once we compiled and verified our preliminary list of dams and reservoirs, we proceeded with the geolocation of each individual record. First, we verified and corrected the data of the preliminary list and then we carried out a second geolocation assessment for our final database using public access online map browsers like Google Earth (https://earth.google.com/web/, last access: 23 May 2020), Bing Maps (https://www.bing.com/maps, last access: 23 May 2020) and Open Street Maps

(https://www.openstreetmap.org/#map, last access: 23 May 2020).

Although these map browsers do not provide us with the analytical capabilities of Geographic Information Systems (GIS) files and programs, these products are operative when visually searching for geographic locations and landmarks, as well as providing data that is often up to date.

In most cases, it was necessary to carry out extensive examinations for each dam since there were cases in which the names of the dams were not sufficient reference to locate them, thus, it was necessary to use additional references such as the nearby cities or villages, the reservoirs names, rivers names, or secondary or alternative names of the dams.

The coordinates in this database are described in decimal degrees using the WGS84 reference coordinate system.

### 2.2.3 HydroSHEDS

To perform the analysis of the dam catchments, the HydroSHEDS (Hydrological data and maps based on SHuttle Elevation Derivatives at multiple Scales) (Lehner et al., 2008) dataset was used. This product allows users access to consistent hydrographic information on a regional scale at a resolution of 15 arc seconds and was derived primarily from the Shuttle Radar Topography    Mission    (SRTM).    The    dataset    information    was    obtained    from    the    public    site

(https://www.hydrosheds.org/downloads, last access: 23 May 2020) in raster format and for this project we utilized 3 layers: void-free elevation, drainage direction and flow accumulation.

Once each dam location was verified and accepted, each location point was aligned according to the HydroSHEDS raster dataset (Lehner et al., 2008) in order to determine the dams' catchments. First, flow direction of each of the model raster cells was computed by applying the 'D8' algorithm. Second, the ridge cells between catchments were identified to delineate them. Finally, the catchment areas were calculated by counting the contributing above cells to each dam.

### 2.2.4 Climatic Research Unit (CRU TS 4.03) time-series dataset

Surface climate variables are commonly used inputs in studies like agriculture, ecology and biodiversity. For this reason, near-surface temperature (NST), precipitation (P) and potential evapotranspiration (PET) mean monthly values from 1901 to 2018 are included for each dam catchment in this database. This data was derived from the Climatic Research Unit (CRU) time-series dataset (Harris et al., 2020), which is hosted by the UK's National Center for Atmospheric Science (NCAS) and it is produced by the University of East Anglia's Climatic Research Unit (CRU). This dataset is a commonly used high-resolution gridded dataset and has been compared favourably with other climatic datasets (Beck et al., 2017; Jacob et al., 2007).

First, the datasets for each variable were downloaded in netCDF formats for monthly periods from 1901 to 2018. Then, these files were converted, resampled and aligned into raster formats in order to match the dams' catchments model. Finally, we computed the long-term mean monthly values for precipitation, near-surface temperature and potential evapotranspiration for the complete time period (1901 to 2018) and for each of the dams' catchments.

This dataset is provided in a resolution of 0.5 degrees by 0.5 degrees grid, it covers the South America continent from 1901 to 2018 and is derived from a periodic interpolation of data from a network of meteorological stations. The NST units are expressed in degrees Celsius (∘C), the PRE units are in expressed in millimetres per month (mm/month) and the PET units are expressed in millimetres per month (mm/month).

For this database we used the version 4.03, which is provided by the Center for Environmental Data Analysis (CEDA) website (https://crudata.uea.ac.uk/cru/data/hrg/#current, last access: 23 May 2020), in a NetCDF format.

### 2.2.5 University of New Hampshire Global Runoff Data Centre (GRDC) composite runoff field

A basic requirement in the assessment of water resource systems is monthly runoff data. For this, the mean monthly runoff data for each dam was also included in this database. We used the University of New Hampshire and Global Runoff Data Centre (UNH/GRDC) Composite Runoff field v1.0 (Fekete et al., 2002), which is often regarded as the best available runoff dataset for large scale models (Gonzàlez-Zeas et al., 2012; Lv et al., 2018). The GRDC dataset combines observed river discharge information with climate-driven water balance models in order to develop consistent composite runoff fields. The method applied in this product uses selected gauging stations data archives to a simulated topological network and compares them with outputs from water balance model (WBM) simulation performed by the authors.

The runoff dataset for South America was downloaded from the data product site in ASCII-grid formats in a resolution of 0.5 degrees by 0.5 degrees. Then, the file was converted, resampled and aligned in order to match the dams' catchments model. Finally, the mean monthly runoff data for each dam catchment was derived. The units of runoff are expressed in millimetres per month (mm/month).

The dataset was obtained from the product public site (http://www.compositerunoff.sr.unh.edu/, last access: 23 May 2020).

**2.2.6 Population data from the Global Rural-urban Mapping Project (GRUMP)**

Demographic data is usually a necessary input for studies that include urban or rural information on water resources assessments. Population for each of the dams' catchments is included on this database and was derived from the Global Rural-urban Mapping Project (GRUMP) (Center for International Earth Science Information Network CIESIN et al., 2011). The GRUMP dataset is provided by the Socioeconomic Data and Applications Center (SEDAC) and offers different georeferenced popula-

tion datasets at continental, regional and national scale. This dataset is often used as baseline for studies that require large-scale maps of urban or rural areas (Florczyk et al., 2020; Mcdonald et al., 2011) and is based on polygons defined by the extent of the night-time light imagery and approximated urban extents from ground-based settlement points.

The dataset was downloaded from the data product public site (https://sedac.ciesin.columbia.edu/data/collection/grump-v1, last access: 23 May 2020) in ASCII format in a 30 arc second resolution. The files were converted, resampled and aligned in

order to match the dam's catchment model, and then the population was computed for each dam catchment. The units of population per dam catchment are expressed in number of people.

**2.2.7 Equipped Area for Irrigation from the Global Map of Irrigated Area dataset**

The equipped area for irrigation (EIA) for each of the dams' catchments were extracted from the Global Map of Irrigated Areas dataset provided by the Food and Agriculture Organization of the United Nations (Siebert et al., 2005) which is often used to

provide valuable information about irrigation in hydrological models (Wisser et al., 2008). This dataset is a global scale dataset of irrigated areas based on cartographic information and FAO statistics and it was developed by combining sub-national irrigation statistics with geospatial information.

The EIA data was downloaded from the data product public site (http://www.fao.org/aquastat/en/geospatial-information/global-maps-irrigated-areas/, last access: 23 May 2020) in ASCII-grid formats, then, the file was converted, resampled

and aligned in order to match the dams catchment model, and then the equipped area for irrigation for each dam catchment was computed. This dataset is presented in a resolution of 0.5 degrees and it is presented in ASCII-grid formats. The units of EIA are expressed in hectares (ha).

### 2.2.8 Aridity Index

The aridity index (AI) is a useful indicator to evaluate long-term climatic water deficiencies on a region. For this study, we
determine the AI for each dam catchment using the methodology proposed by UNESCO (UNEP et al., 1992) which is represented by:

$$AI_i = \frac{P}{PET} \qquad [1]$$

Where $AI_i$ is the aridity index for each dam catchment, $P$ is the mean annual value of precipitation for each dam catchment (mm/year) and $PET$ is the mean annual potential evapotranspiration for each dam catchment (mm/year). The aridity index is unitless. Both the mean annual precipitation and potential evapotranspiration values are derived from the CRU dataset. The
units for both P and PET values are expressed in millimetres per year.

In general, higher values of AI represent humid climates, while lower values represent dry or arid climates. Aridity indexes are commonly classified based on the following subtypes: hyper-arid (AI<0.03), arid (0.03≤AI<0.20), semi-arid (0.20≤AI<0.50), subhumid, (0.50≤AI<0.65) and humid (AI≥0.65) (Pour et al., 2020).

### 2.2.9 Residence Time

The residence time (RT) or the 'age' of water, is a common indicator used to determine useful information about the storage, sediment transport, water quality or flow pathways of a catchment (Mcguire et al., 2005; Vörösmarty et al., 2003). This indicator usually refers to local conditions in a single reservoir and is usually represented by:

$$RT_i = \frac{reservoir\ volume_i}{discharge\ volume} \qquad [2]$$

Where $RT_i$ is the residence time for each reservoir, *reservoir volume$_i$* is the volume of the reservoir $i$, and *discharge volume* is the average discharge volume per year at each dam $i$. If reservoir volume is expressed in cubic kilometres and discharge volume
is expressed in cubic kilometres per year, residence time is expressed in years.

For the annual discharge volume, we used the information from the GRDC composite runoff field dataset and the area of each dam catchment which was derived from the HydroSHEDS dataset.

### 2.2.10 Degree of Regulation

The degree of regulation (DOR) provides a first approach to assess the potential impact of reservoirs on their downstream
network. This index measures the degree of flow regulation that a dam or a cluster of dams can cause on a river network. This regulation alters the connectivity of the streams and can cause disruptions on seasonal flow events or can reduce the transport of sediments or species though the river network (Grill et al., 2019; Lehner et al., 2011).

In order to determine the DOR index, we followed the methodology described by (Grill et al., 2019) and computed the DOR index for each dam location based on the relationship between the accumulated reservoir volume and the total annual flow
river at each dam's location. This index is determined in percentage and is represented by:

$$DOR_i = \frac{\sum_{j=1}^{n} reservoir\ volume_j}{discharge\ volume} \qquad [3]$$

Where $DOR_i$ is the degree of regulation index for each stream reach $i$, *reservoir volume$_j$* is the reservoir volume of the dams $j$ located upstream or the stream reach $i$, $n$ is the total number of upstream dams, and *discharge volume* is the average discharge volume per year at the stream reach $i$. For this study we used a minimum threshold of 2% to distinguish between free-flowing rivers  (Dynesius and Nilsson, 1994) and also, we restricted the DOR value to 100% to limit multi-year reservoirs to the same maximum DOR (Lehner et al., 2011).

We extracted the river network from the HydroSHEDS dataset and defined the rivers as the streams that exceeded an upstream catchment area of 10 km$^2$. For the annual discharge volume, we used the information from the GRDC composite runoff field dataset and the area of each dam catchment which was derived from the HydroSHEDS dataset. Reservoir volume is expressed in cubic kilometres and the discharge volume is expressed in cubic kilometres per year. The degree of regulation is expressed in percentage values.

## 3 Results

### 3.1 Dams and Reservoirs

Once the review, refinement and processing of the data was concluded, a total of 1,010 dam entries were accepted for our database (Figure 1). This represents a noticeable progress in the identification and geolocation of dams in the region and thus, enables the opportunity for new research that allows a more precise understanding of the water resources systems in the region. After a comparison with other databases, 376 entries were similar to the AQUASTAT and GRaND databases; however, they were included in our database since the 1,010 entries were inspected and verified following the same procedure described in previous sections. Additionally, this database increases dam entries not only as a total regional number but also increases the number of entries per country, which means that with this database we also expect to contribute to new research in study areas that have not been considered to date due to the absence of reliable information. Table 2 details the entries in our database for each country considered in this study, including a comparison with the AQUASTAT and GRaND databases. Table 3 describes the 24 variables processed and accepted for this database. The estimated total reservoir volume of this database is 1,017 cubic kilometres and the largest reservoir belongs to the "El Guri" dam in Venezuela with an estimated volume of 135 cubic kilometres.

We also present an analysis on the implementation of dams in South America. This analysis is shown in Figures 2a and 2b. Our results show that the largest number of dams were built since the 1960s, a period in which more than 70% of the dams on the continent have been built. Similarly, the greatest increase in storage capacity occurred between the 1970s and the 1990s, which suggests that the largest projects were implemented in this period, including the "El Guri" dam. In the case of dams implemented by countries, we can observe the relevance of Brazil, the country with the highest number of dams in our database

with more than 50% of records. This predominance is also seen in the total storage volume, since Brazil has more than 60% of the total volume of storage reported in our database, probably due to the vast amount of water resources in this country.

## 3.2 Hydrological Information

The model derived from the HydroSHEDS dataset allowed us to determine the catchment areas of this database, which were necessary to carry out the subsequent hydrological calculations. The accumulated area of the dams' catchments is approxi-

mately 14,855,192 km$^2$ with an average catchment of 18,385 km$^2$. The largest catchment belongs to the "Jirau" dam in Brazil with an estimated area of 962,732 km$^2$. Table 4 describes the variables processed for the hydrological information included in this database.

Figure 3 presents the annual values for NST, P, PET and runoff estimated for each dam catchment. Both in the case of NST and P, higher values would seem to be mostly located near the equator, while PET higher values are more noticeable in the

northeast of Brazil. In the case of runoff, values are scattered and there is no evident predominance, except for higher values localized in the southeast of Brazil.

Figures 2c and 4a represent the values of catchment population per each dam catchment. We observe a clear connection between these attributes, with larger catchment areas corresponding to larger populations. Although this trend by itself is expected, figure 4a suggests a strong population pressure on downstream catchments, which is mainly inflicted by upstream

population catchments. For example, the "Yacyretá" dam has the largest population with more than 55 million people. However, this value comes mainly due to the accumulated population of upstream catchments, including the "Itaipú" dam catchment population of almost 49 million people, which in turn also receives most of its large catchment population from upstream catchments. Figure 4b presents the equipped area for irrigation for each dam catchment. The dam with the largest equipped area for irrigation corresponds to "Yacyretá" dam catchment dam with more than 930,000 hectares of equipped areas for

irrigation.

Figure 2d and 4c describe the number of dams per aridity index type and per country. In this case, we observe that dams located in arid areas are mostly located in the southwest of the continent, especially in Argentina, Chile and Peru. Most significantly, we observe that two dams: 'Austral' and 'Candelaria' have their catchments located in hyper-arid areas. In the case of catchments located in humid areas, we observe that most of these dams are located near the equator, largely due to the high precip-

itation values in this region.

Figure 2e describes the relationship of runoff and residence time per dam. We observe a clear relation between these two attributes, with reservoirs with larger specific capacity corresponding to catchments with lower runoff values. This indicates an 'expected' performance from most of the dams in our database, from large reservoirs located in regions with low available water resources areas like the 'Cocorobó' dam in Brazil or the 'Las Maderas' dam in Argentina. In the opposite side, we

observe small reservoirs in large water resources areas like the 'Chisaca' dam in Colombia or the 'Suytococha' dam in Peru. Figure 4d describes the residence time for each dam. Again, if we visually compare this figure with figure 3d, we observe a clear relation between the residence time and runoff, with high residence time values located in areas with low runoff areas.

Finally, figure 2f provides information on the DOR index in the rivers of South America classified by river flow category and level of regulation. The river flow category refers to different values of average mean flow. Our results indicate that the regulation effects of reservoirs are more evident in the rivers with smaller average flows. Over 50% of the total "affected rivers" in the region, these are the rivers with a DOR>=2%, correspond to small flow rivers. The DOR affectation decreases as the mean river flow increases, which is observed in very large average flows, whose level of DOR affectation is less than 1%. Rivers with multi annual reservoirs, this is streams with a DOR=100%, are more frequent in small flow rivers, with more than 27% of the total observations. Figure 5 shows the degree of river regulation of the reservoirs of the DDSA database for the "affected" rivers of South America.

## 4 Data limitations and uncertainties

The information provided in this database cannot be considered error free since it has been prepared using the information available at the time of its elaboration. It should also be noted that although our database was created independently, through an individual investigation and based primarily on reports and documents available from each of the countries in the region, the database may include attributes of dams that are also reported by other existing dam databases such as ICOLD, AQUASTAT and GRaND.

Hydrological inputs provided in this database also need careful interpretation to avoid misleading interpretations. First, the resolution of the hydrological datasets used in the DDSA database could affect the accuracy of results for small catchments. Although all the datasets considered in this database have been largely validated for large-scale or regional assessments models (Gonzàlez-Zeas et al., 2012; Lv et al., 2018), we suggest caution if the intention is to use these results in catchments with an area smaller than the cell size of each dataset.

Our results regarding aridity index, residence time and degree of regulation also need to be interpreted with caution. First, our results are intended to assess the dams' catchments and therefore, should be used carefully if intended for other type of assessment. Also, in the case of the DOR index, there are many important inputs in our assessment which have not been considered and may alter the assessment results. For example, given the scale of this study, we are not considering information about local water use, specific stream characteristics or relevant and updated urban information. Also, two relevant inputs were not considered in our DOR assessment: unidentified small reservoirs, and the reservoir's active storage instead of total reservoir storage. These inputs should be considered in order to obtain more accurate results of the flow regulation but were not considered due to the absence of this information. Furthermore, the impacts of river regulation also depend on a wide range of factors, e.g. local or international water management policies, which have not been considered either. Altogether, we consider that despite the aforementioned uncertainty factors, our results give a consistent first approximation of these indices at a regional scale.

Finally, in order to assess the robustness of our DOR assessment, we conducted a sensitivity analysis by comparing our findings with the results determined by (Grill et al., 2019) in their manuscript 'Mapping the world's free-flowing rivers' (DoR_FFR).

Figure 6 compares 409 stream matches from both studies and determines a strong correlation (r=0.702) between our results and the DoR_FFR manuscript. The correlation results are more evident on large and very large rivers.

## 5 Data availability

The Database of Georeferenced Dams of South America (DDSA) is a joint effort of researchers from the Department of Civil Engineering: Hydraulics, Energy and Environment of the Universidad Politécnica de Madrid and the Civil Engineering Career

of the Universidad Técnica de Ambato. The DDSA database is available for both researchers and the general public through the ZENODO open access repository https://doi.org/10.5281/zenodo.4315647 (Paredes-Beltran et al., 2020), where we have detailed the contact information of the authors, in order to receive any valuable contribution which could allow us to improve our database.

## 6 Summary

The database of georeferenced dams in South America (DDSA) has been developed to contribute to the improvement of water resources management in the region. The provision of reliable, high-resolution and available data on dams and reservoirs will contribute in the assessment of freshwater ecosystems and communities both for present and future scenarios in this region, which to this date, have been restricted to a limited number of catchments due to the absence of available information, and thus, contributing to generate more informed decision-making processes in order to safeguard the future sustainability of the

communities in this region.

The 1,010 entries of dams present a total of 24 attributes. Each record has been included in the list after an individual review and its position has been determined considering public digital terrain models. In addition, the database also provides mean monthly hydrological information. With this increased spatial coverage and attributes information, this database could be used as a baseline for further studies that address relevant issues regarding dams, hydrology, ecology and people in the region. Also,

with the inclusion of data for all the countries in the continent we also expect to contribute to an in-depth comprehension on the hydrological and environmental dynamics for the entire continent, and encouraging the generation of knowledge in areas that have not been considered in past studies.

One of the main goals of this endeavour is to foster the research of water resources in South America. To achieve this objective, we consider that we must make the necessary efforts to keep our database relevant to the international hydrology community.

For this, we believe it will be necessary to keep our database updated, and also, include additional information regarding hydrology and water resources management in future versions of our database. Future dams are one of the topics we need to observe to maintain our database updated. In recent years, several South American countries have made public their intention to develop new dam projects, mainly for hydroelectric generation (Almeida et al., 2019; Anderson et al., 2018; Moran et al., 2018; Zhang et al., 2018). We have identified 574 future projects in South America, 61 under construction for 2020 and 513

projects planned to be developed in the future. Supplementary Table 1 details future dams in South America identified by country, name and implementation phase.

Monitoring the development of future dams in South America is necessary due to the relevance of these projects on the local and regional scales. It is not likely that all projects listed in Supplementary Table 1 will be carried out due to different economic, social or political factors (Anderson et al., 2018). However, the likely ecological or social impacts that these projects may cause (Doria et al., 2018; Lees et al., 2016; Winemiller et al., 2016) highlight the necessity for the international hydrological community to be conscious of the status of these projects.

Similarly, we consider that future versions of our database may be extended with additional attributes. For example, information such as outflows of dams (discharge time series) or energy generation data from hydroelectric dams (energy generation time series), could also be included in the future. However, to date, including this type of information on a continental scale represents a significantly great effort due to the lack of readily available information on water resources in most countries of the region. There are countries, like Brazil, which make public their relevant information about water resources and energy generation through their official agencies, e.g. the National Agency of Water ANA (https://www.ana.gov.br/sar/sin, last access: 9 Nov 2020), and the National Electric Energy Agency ANEEL (https://www.aneel.gov.br/siga, last access: 9 Nov 2020). Then again, other countries of the region keep this information restricted or outdated, which makes it difficult to complete these attributes for the entire database.

Finally, the data presented in this database is largely based on open-access information available to date, therefore, the valuable support of both public institutions and the international hydrology community will be necessary for extending future versions of our database. This will allow us to keep our database relevant, which in turn will support the development of future research initiatives on water resources in the region.

**Acknowledgments**

The first author would like to acknowledge Universidad Técnica de Ambato for the financial support through its doctoral student mobility program (award no. 1886-CU-P-2018 Resolución HCU). The authors acknowledge the financial support of the Ministerio de Ciencia e Innovación (Spain) through the project SECA-SRH (PID2019-105852RA-I00). The authors thank two anonymous reviewers and editor David Carlson for their valuable feedback on our paper.

**Author contributions**

BEPB: Conceptualization, Investigation, Data curation, Validation, Software, Formal analysis, Methodology, Writing - original draft, Writing - review & editing. ASW: Conceptualization, Formal analysis, Funding acquisition, Methodology, Validation, Writing - review & editing, Supervision, Resources. LG: Formal analysis, Software, Funding acquisition, Validation, Writing - review & editing, Supervision, Resources, Project administration.

## Competing interests

The authors declare that they have no conflict of interest.

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

**Table 1:** Available public data records of dams per country

| COUNTRY | AVAILABLE PUBLIC INFORMATION | NUMBER OF ENTRIES | GEOREFERENCED INFORMATION | REFERENCE* |
|---|---|---|---|---|
| ARGENTINA | Inventario de Presas y Centrales Hidroeléctricas de la República Argentina | 31 | No | (Subsecretaría de Recursos Hídricos, 2010) |
| BOLIVIA | Inventario Nacional de Presas Bolivia | 287 | Yes | (Programa de Desarrollo Agropecuario Sustentable (PROAGRO), 2010) |
| BRAZIL | Mapa Interativo das Barragens Cadastradas | 18,880 | Yes | (Agencia Nacional de Aguas, 2019) |
| CHILE | Directorio de Presas | 107 | No | (Comite Nacional Chileno de Grandes Presas, 2019) |
| COLOMBIA | ISAGEN | - | No | (ISAGEN, 2020) |
| ECUADOR | Various web pages | - | No | (ElecAustro, 2020; Hidroabanico, 2019; Instituto Nacional de Pesca, 2020) |
| FRENCH GUIANA | Not available | - | - | |
| GUYANA | Not available | - | - | |
| PARAGUAY | Not available | - | - | |
| PERÚ | Inventario de Presas en el Perú | 743 | Yes | (Autoridad Nacional del Agua, 2016) |
| SURINAME | Not available | - | - | |
| URUGUAY | Not available | - | - | |
| VENEZUELA | Other | - | No | (Instituto Nacional de Estadistica, 2020) |

* The data records of each country website links are detailed in the reference section

**Table 2:** Number of new dam entries per country

| COUNTRY | DDSA | OTHER DATABASES | | | NEW ENTRIES PER COUNTRY |
|---|---|---|---|---|---|
| | | GRAND | AQUASTAT | SIMILAR ENTRIES TO DDSA* | |
| ARGENTINA | 107 | 35 | 0 | 35 | 72 |
| BOLIVIA | 66 | 3 | 56 | 57 | 9 |
| BRAZIL | 507 | 182 | 0 | 182 | 325 |
| CHILE | 73 | 10 | 0 | 10 | 63 |
| COLOMBIA | 58 | 24 | 16 | 33 | 25 |
| ECUADOR | 21 | 6 | 2 | 6 | 15 |
| FRENCH GUIANA | 2 | 1 | 1 | 1 | 1 |
| GUYANA | 3 | 0 | 0 | 0 | 3 |
| PARAGUAY | 4 | 2 | 0 | 2 | 2 |
| PERÚ | 73 | 13 | 0 | 13 | 60 |
| SURIMANE | 1 | 1 | 0 | 1 | 0 |
| URUGUAY | 13 | 4 | 0 | 4 | 9 |
| VENEZUELA | 82 | 32 | 0 | 32 | 50 |
| **TOTAL** | **1,010** | **313** | **75** | **376** | **634** |

* In some cases, AQUASTAT and GRaND entries were duplicated, so they were considered as a single entry

**Table 3:** List of variables processed for dams and reservoirs

| *VARIABLE* | UNIT | DESCRIPTION |
|---|---|---|
| **ID** | | Unique Id number for each dam |
| **NAME OF THE DAM** | | Name of the dam |
| **OTHER NAME** | | Alternative names given to the dam (aliases, former names) |
| **DECIMAL DEGREE LATITUDE** | Decimal Degrees | Latitude coordinate of point location of the dam. |
| **DECIMAL DEGREE LONGITUDE** | Decimal Degrees | Longitude coordinate of point location of the dam |
| **HEIGHT** | m | Height of the dam above foundation expressed in meters |
| **LENGTH** | m | Length of the dam measured at the crest expressed in meters |
| **RESERVOIR CAPACITY** | MCM | Capacity of the reservoir expressed in million cubic meters |
| **RESERVOIR AREA** | $km^2$ | Area of the reservoir expressed in square kilometres |
| **RESERVOIR NAME** | | Name of the reservoir or water body, if different from the dam name |
| **RIVER** | | Name of the river in which the dam is located |
| **INTERNATIONAL** | | Indicates if the dams or reservoirs lie within more than one country |
| **YEAR OF COMPLETION** | | Reported year of completion of the dam |
| **FLOOD CONTROL** | | Use of the dam for Flood Control |
| **IRRIGATION** | | Use of the dam for Irrigation |
| **HYDROELECTRICITY** | | Use of the dam for Hydroelectricity |
| **NAVIGATION** | | Use of the dam for Navigation |
| **RECREATION** | | Use of the dam for Recreation |
| **WATER SUPPLY** | | Use of the dam for Water Supply |
| **OTHER USE** | | Use of the dam for other purposes |
| **COUNTRY** | | Name of country |
| **NEAREST TOWN** | | Name of the nearest town or city to the dam location |
| **STATE / PROVINCE** | | Additional information about the location of the dam |
| **NOTE** | | Specific comments of importance |

**Table 4:** List of hydrological and additional information processed in this study

| *VARIABLE* | UNIT | DESCRIPTION |
| --- | --- | --- |
| **CATCHMENT AREA** | km$^2$ | Calculated catchment area per dam expressed in square kilometres |
| **NEAR SURFACE TEMPERATURE** | $\circ$C | Calculated monthly average near surface temperature value derived from the Climatic Research Unit (CRU TS 4.03) time-series dataset per each dam catchment expressed in degrees Celsius |
| **PRECIPITATION** | mm/month | Calculated monthly average precipitation value derived from the Climatic Research Unit (CRU TS 4.03) time-series dataset per each dam catchment expressed in millimetres per month |
| **POTENTIAL EVAPOTRANSPIRATION** | mm/day | Calculated monthly average potential evapotranspiration value calculated using the Pen-Monteith method derived from the Climatic Research Unit (CRU TS 4.03) time-series dataset per each dam catchment expressed in millimetres per day |
| **GRDC** | mm/month | Calculated monthly average monthly runoff derived from the University of New Hampshire Global Runoff Data Centre (GRDC) composite runoff field per each dam catchment expressed in millimetres per month |
| **POPULATION** | people | Calculated population data from the Global Rural-urban Mapping Project (GRUMP) per dam catchment |
| **IRRIGATION** | ha | Calculated irrigation area from the Global Map of Irrigated Area dataset per dam catchment expressed in hectares. |
| **ARIDITY INDEX** | n/a | Calculated aridity index per dam catchment. The aridity index is unitless. |
| **RESIDENCE TIME** | years | Calculated residence time index per each dam reservoir. The residence time is expressed in years. |
| **DEGREE OF REGULATION** | % | Calculated degree of regulation index per dam stream reach. Degree of regulation is expressed in percentage. |


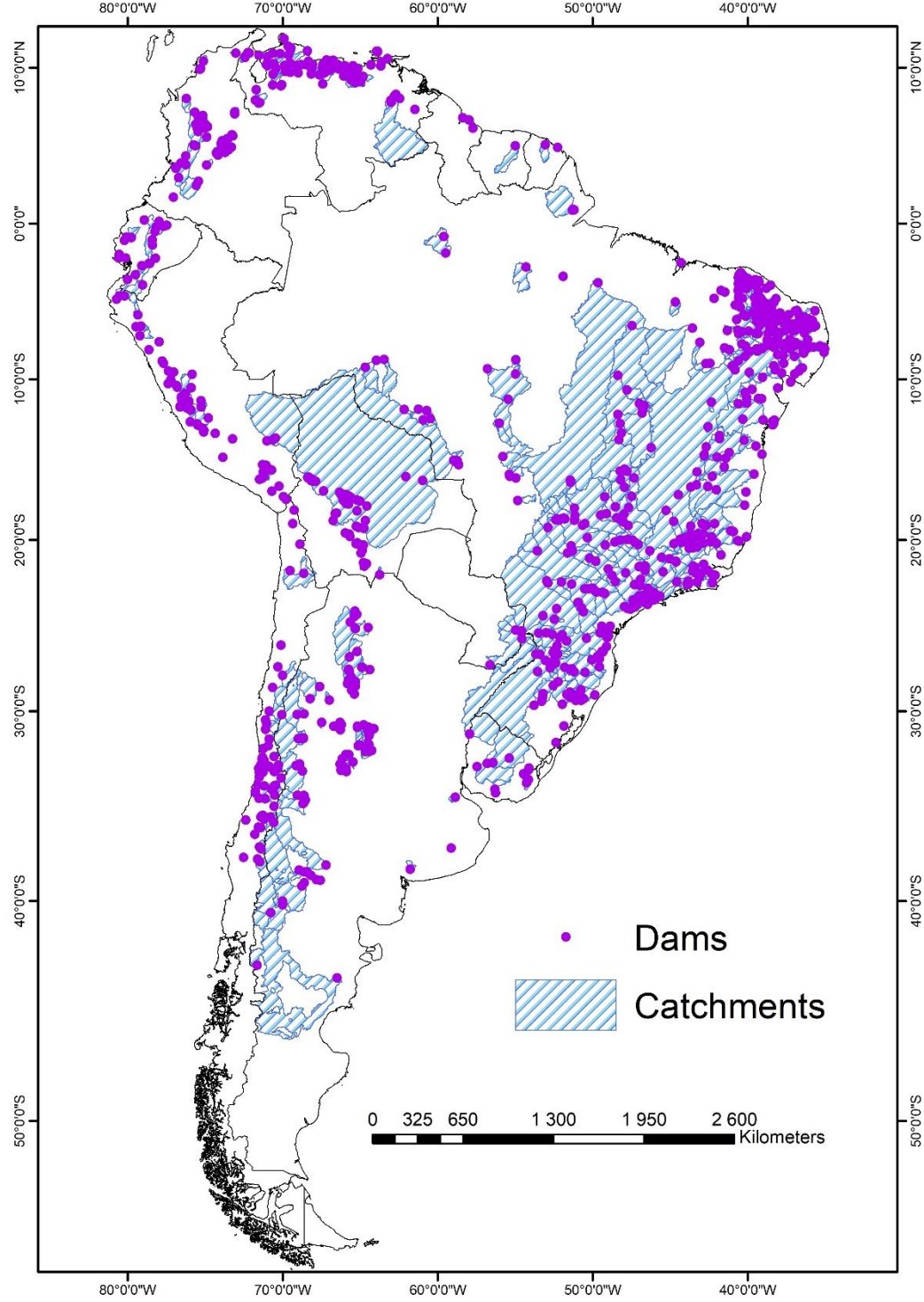

**Figure 1.** Dataset of Georeferenced Dams in South America (DDSA)

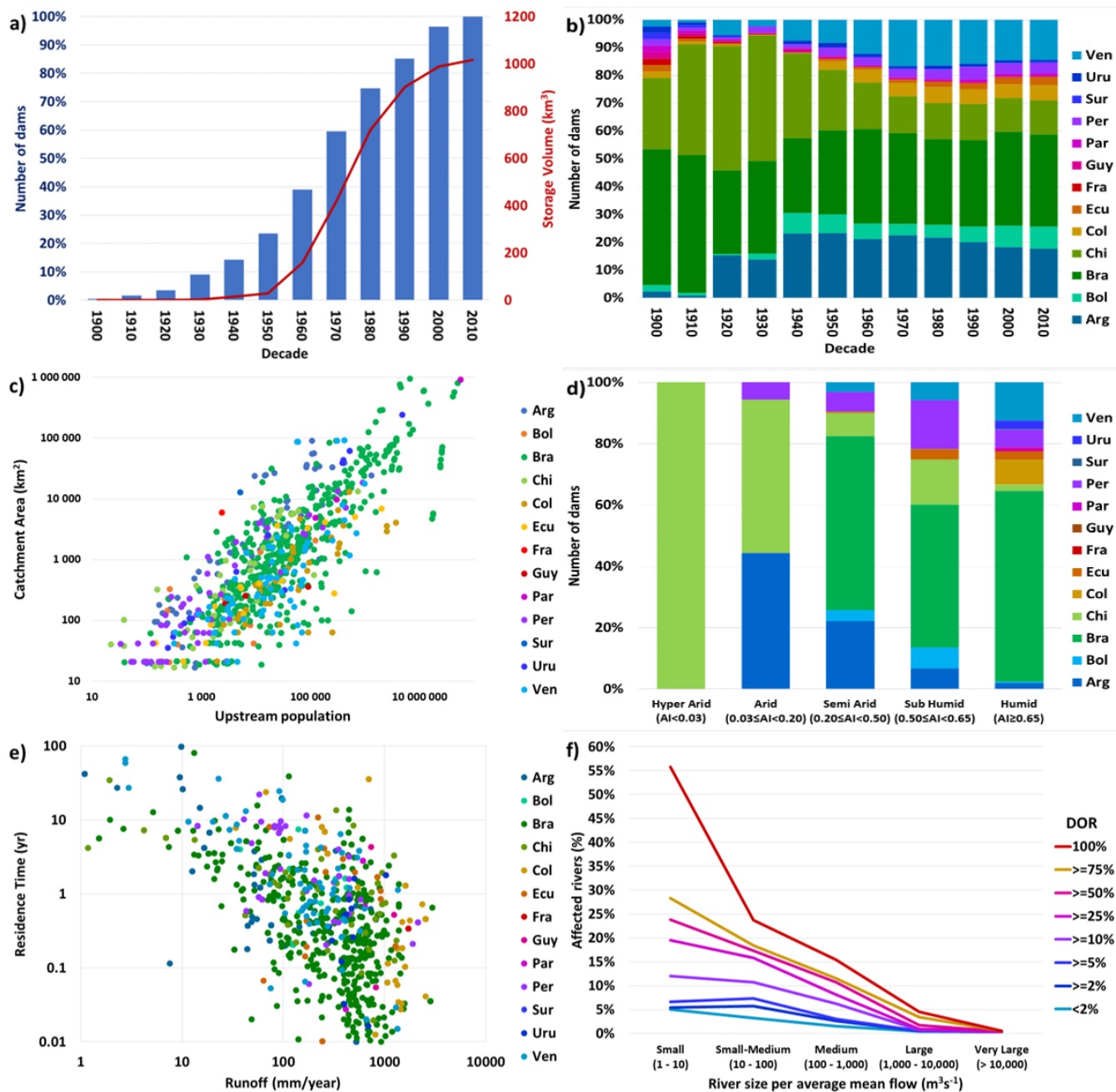

**Figure 2 a)** Cumulative number of dams per decade and per storage volume **b)** Cumulative number of dams per decade per country **c)** Upstream population per catchment area and per country **d)** Number of dams per aridity index type and per country **e)** Annual catchment runoff per residence time and per country **f)** Cumulative affected rivers per river size and per different DOR range

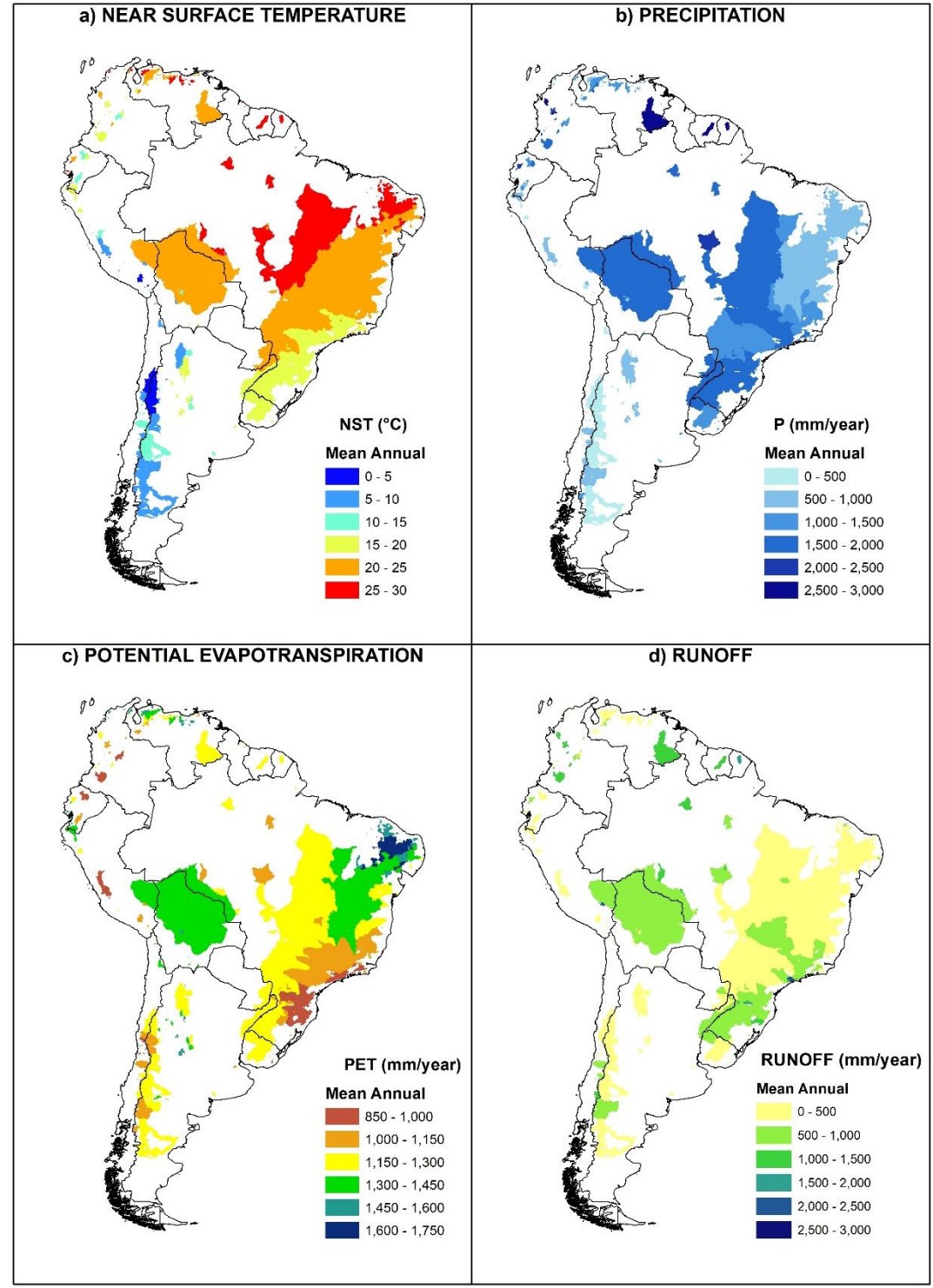


**Figure 3. a)** Near Surface Temperature**, b)** Precipitation**, c)** Potential Evapotranspiration**, d)** Runoff maps per dam catchment

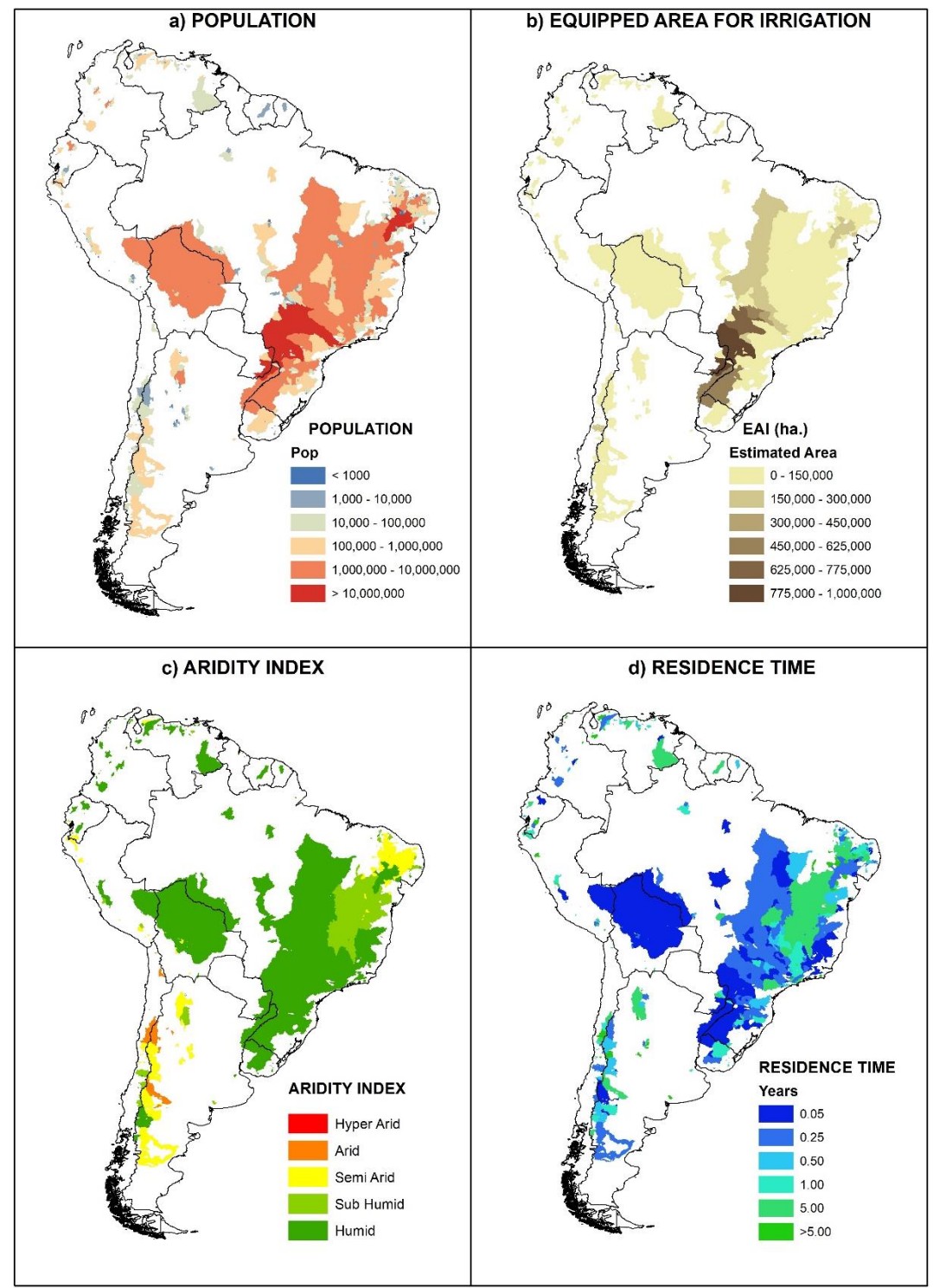

**Figure 4. a)** Population, **b)** Equipped Area for Irrigation, **c)** Aridity Index, **d)** Residence Time per dam catchment.

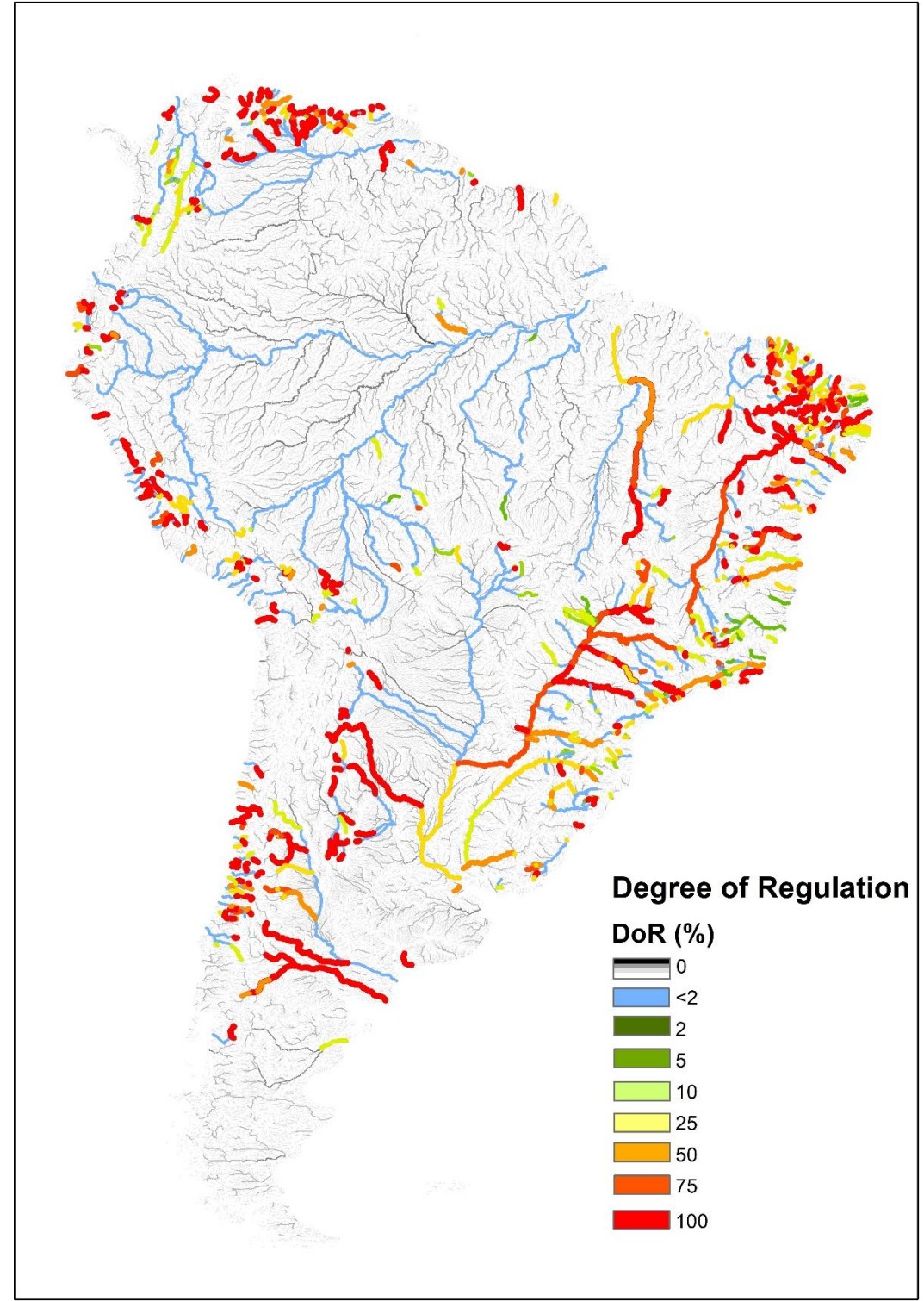

**Figure 5** Degree of Regulation (DoR) of reservoirs of the DDSA database in downstream rivers of South America.

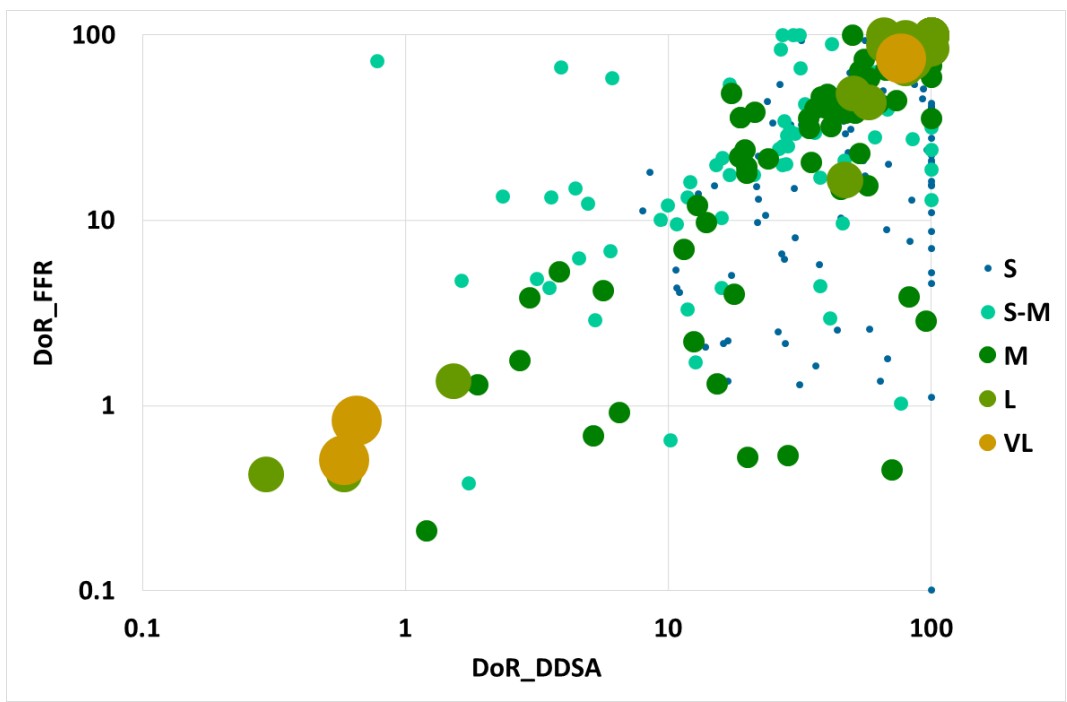

**Figure 6** DOR sensitivity analysis. Degree of regulation values from DDSA database (DoR_DDSA) were compared with matching values from (Grill et al., 2019) degree of regulation values (DoR_FFR). River were classified by their average mean flow; smaller dots represent small rivers and bigger dots represent large to very large rivers.