# Peer review of "Dataset of Georeferenced Dams in South America (DDSA)"

_Earth System Science Data, 2020_

## Referee Comment (RC1) · Anonymous Referee #1 · 21 Sep 2020

The manuscript 'Dataset of Georeferenced Dams in South America (DDSA)' presents a very important compilation of georeferenced dams in South America (SA). Since most global databases do not include many important dams in SA, it is indeed paramount that regional initiatives as the one presented here be developed to foster water management in the continent. I thus support the publication of this manuscript in ESSD, after some revisions as highlighted below, and for this I suggest major revisions.

Firstly, a section with perspectives for future developments of large scale datasets of dams in SA could be included. For example, this dataset provides mainly information on the location of the dams. However, other data are also fundamental to foster water management across the continent, e.g., availability of dam outflows (i.e. discharge time series). For instance, Brazil's ONS (Operador Nacional do Sistema) provides daily

discharge data and reservoir storage for most reservoirs in the national interconnected system (https://www.ana.gov.br/sar/sin). These data were used for example for a national scale assessment by Passaia et al 2020 (Impact of large reservoirs on simulated discharges of Brazilian rivers; Brazilian Journal of Water Resources). Another information relates to time series of energy generation, and some SA countries also make it available online (e.g., Brazil, Colombia). I think a paragraph could be included to discuss which kind of information would be interesting for improving water management related to reservoirs in SA (and which datasets already exist and are not included in DDSA). This could push the international hydrology community somehow to develop new initiatives of data sharing.

Future dams (i.e. proposed dams or dams under construction) are also neither included nor discussed in the text. I think it should be included somehow (at least a paragraph about it). For instance, ANEEL (Brazilian energy agency) has an available shapefile of the status of dams in the country (in operation, proposed, at inventory phase, etc). The FHReD dataset also provides proposed dams worldwide, which includes many in SA (http://globaldamwatch.org/fhred/).

The authors could consider presenting an updated map of the degree of regulation index (DoR; basically the total storage of upstream reservoirs divided by the average discharge at a given river reach) which is a simple one yet powerful to understand reservoir regulation at large drainage networks. This is easy to do, since the authors already have the Hydrosheds ID for each dam location. This would be a kind of updating for SA of the free-flowing rivers map published recently (Grill et al 2019 Nature).

The interpretation of the hydrological data and the outcomes of the dataset in the Results section is too simplistic. For example, in the section 3.1 Dams and Reservoirs there is only a comparison with GRanD and AQUASTAST databases. However, given the large amount of data available, more interesting figures as histograms with number of dams implemented per year and per country should be included. Regional analyses could also be performed, e.g., higher dams are mainly located in which countries, in
which type of environment? Although I recognize that this is mainly a paper describing the dataset itself, some additional analyses could be included and would certainly improve the overall quality of the manuscript. In section 3.2 Hydrological Information, the authors focus on describing extreme values of PET, Precipitation, temperature and other variables at individual sites (e.g., 'The highest potential evapotranspiration record is documented for the catchment of the "Pilões" dam in Brazil with 1,713.32 millimetres per year'). However, for a continental scale dataset as this one, I think that regional analyses would be much more interesting, e.g., how many dams are located in regions with high aridity index (PET/P)? Similarly, in section '3.3 Additional Information', there is only a simple phrase on how Yaciretá dam is associated to the highest upstream population and equipped areas of irrigation. A more thorough analysis describing the distribution of dams at different levels of population pressure across the continent could be included.

The authors could consider analyzing upstream population divided by the dam drainage area, this would put some weight on the large upstream population for dams located in downstream reaches as Yaciretá dam in the Paraná river.

Why is 'equipped areas of irrigation' considered an 'additional information'? For me it is a hydrological information.

More information on the data used (section 2.2) should be provided. For example, some information is missing, as the unity of catchment irrigation area (this is only presented in figure 3, and it not presented in the main manuscript or in the provided data in Zenodo).

The authors use the catchments of each dam to estimate some properties (upstream population, etc). The catchment polygons are presented in Figures 2 and 3. I think a shapefile with the polygons should also be provided in the Zenodo dataset, what is very useful for users to extract other interesting information, and it would be in the context of other initiatives of hydrological datasets as CAMELS-Chile (Alvarez-Garreton et al

2018 HESS) and CAMELS-Brazil (Chagas et al 2020 ESSD).

Finally, some text clarifications are still required in some parts. Some paragraphs are also too long and must be reduced or splitted. I provide some minor suggestions below.

Minor suggestions:

Line 9 Split into two sentences: 'In general, its relevance relies on facilitating the management of water resources for anthropogenic purposes. However, dams could also generate many potential adverse impacts related to safety, ecology or biodiversity.'

L.18 'dams' catchments'

L.23 'contribute to the development...'

L.33 'assess'

L.49 'La Plata' instead of 'El Plata'

L.52 'which reports' instead of 'and reports'

L.54 '...America it only reports...'

L.72 check: '5,283,000'

L.74 'Paraná' with acute accent

L.81 '...the continent, there exist humid...'

L.84 'found' instead of 'find'

L.85 '...Chile, which are blocked due to the Andes mountains, which causes low precipitation...'

L.88 'and it is located'

L.89 'for example the "El Niño",...'

L.89-90 this whole phrase is confusing, please re-phrase. Besides, it is too simplistic

to state that ENSO 'increases precipitation at the northwest area', since it affects in very different ways different regions of South America. Please improve this description here.

L.96 'The GRanD' - 'The' should be in lowercase.

L.112 a reference for HydroSHEDS should be included (Lehner et al), not only the dataset website

L.119 a reference for CRU should be included (New et al)

L.127 a reference for GRDC should be included

L.135 a reference for GRUMP should be included

L.135 'for each of dams' catchments...'

L.141 'catchment were extracted'

L.180 'reservoirs' catchments' instead of 'reservoir's catchments': please check this throughout the whole text. The 'catchments' refer to all 'reservoirs', and not just to one reservoir and stated in the current form 'reservoir's catchments'

L.195 'performed' instead of 'calculated'

L.195 which statistical analysis was performed? or was it just a long term average for each month?

L.225 '...of the data was concluded, ...'

L.225-226 phrase too long, please reduce it or split into two phrases.

L.228 'GRanD' instead of 'GrAND' - please check throughout the text

L.240 '14,855,192'

L.240 please split phrase in two: '...kilometres. The largest catchments... '

L.242 'Our results highlight the great influence and importance of the Amazon rainforest in the continent since most of the highest records': I don't understand the relevance of this phrase for the context of a database of dams.

L.248 this runoff value of 2961 mm/year for Billings catchment is certainly a model error, since it does not rain that much in this catchment to have this runoff. The high precipitation rates occur more in the mountains close to São Paulo. You can check it in the Brazilian precipitation maps by the Brazilian Geological Survey (CPRM): <http://www.cprm.gov.br/publique/Hidrologia/Mapas-e-Publicacoes/Atlas-Pluviometrico-do-Brasil-1351.html> The runoff model uncertainty should be discussed here. I honestly

L.450 I don't understand why Figure 3 has figures e) and f), and not a) and b), since it is a figure by itself, and not a continuation of Fig 2.

---

## Referee Comment (RC2) · Anonymous Referee #2 · 23 Sep 2020

This manuscript entitled "Dataset of Georeferenced Dams in South America (DDSA)" clearly explains the importance of a database of georeferenced dams in South America, along with other variables that describe each local dam scenario. Thus, the purpose of the proposed work is clear and important in order to establish a base line of information for other studies related to dams and reservoirs. I think the idea and work are in concordance with the journal's and I encourage the authors to modify the text and also include guidelines for future database modifications (i.e. updates). I recommend this article for publication upon the following major revisions:

The relevance of the proposed database for future studies will be directly linked with latest update. Thus, a mechanism for complementing the information should be provided. £What should other researchers do in order to update the data on a given

region?

The Data description section is divided in three subsections: 2.1. Study Area. 2.2. Data Sources, and 2.3. Data Processing. Subsections 2.2 and 2.3 repeat a significant portion of the information. I would suggest to the authors to fusion these two into a single subsection "Data sources and assessments methods" maintaining the 7 groups originally indicated in subsection 2.2 and including within each of them the methods used for the respective data assessment.

Line 25. Mentions that dams are, in many cases, controversial due to "some associated negative impacts..." and the following sentence indicates that "these structures cause major impacts and changes wherever these are implemented"- this is ambiguous. Please refer to the impacts complementing line 40 in which a list of them is considered and complement it with other topics such as, morphology, water quality and habitat. Please classify them as acute and chronic impacts (time related impacts) since it will add value to the continuous monitoring efforts.

Line 65. Please explain which "fields" the authors are referring to.

Line 70 and 75 mention the Amazon, Parana-Rio de la Plata, and the Orinoco rivers as the largest systems in the region. Please revise to consider mentioning these fluvial systems only once when describing the study area.

Line 85 refers to "El Niño" however there is no mention to "La Niña" that also brings changes to the precipitation patterns and it should be considered.

Line 100 cites Table 1 regarding the government's available information. Please include the respective links to the information. This is important in order to replicate efforts in the future.

―――――――――――――――――――

---

## Author Comment (AC1) · 20 Nov 2020

**Response to Anonymous Referee #1**

*1.*  The manuscript 'Dataset of Georeferenced Dams in South America (DDSA)' presents a very important compilation of georeferenced dams in South America (SA). Since most global databases do not include many important dams in SA, it is indeed paramount that regional initiatives as the one presented here be developed to foster water management in the continent. I thus support the publication of this manuscript in ESSD, after some revisions as highlighted below, and for this I suggest major revisions

*Response:* The authors take the opportunity to acknowledge the valuable comments provided by the anonymous referee, as well as the time that has been committed to provide this valuable feedback. All suggestions made have been considered and addressed in a reasoned manner. Revisions have been made to the manuscript and are described below.

*2.*  Firstly, a section with perspectives for future developments of large-scale datasets of dams in SA could be included. For example, this dataset provides mainly information on the location of the dams. However, other data are also fundamental to foster water management across the continent, e.g., availability of dam outflows (i.e. discharge time series). For instance, Brazil's ONS (Operador Nacional do Sistema) provides daily discharge data and reservoir storage for most reservoirs in the national interconnected system (https://www.ana.gov.br/sar/sin). These data were used for example for a national scale assessment by Passaia et al 2020 (Impact of large reservoirs on simulated discharges of Brazilian rivers; Brazilian Journal of Water Resources). Another information relates to time series of energy generation, and some SA countries also make it available online (e.g., Brazil, Colombia). I think a paragraph could be included to discuss which kind of information would be interesting for improving water management related to reservoirs in SA (and which datasets already exist and are not included in DDSA). This could push the international hydrology community somehow to develop new initiatives of data sharing.

Future dams (i.e. proposed dams or dams under construction) are also neither included nor discussed in the text. I think it should be included somehow (at least a paragraph about it). For instance, ANEEL (Brazilian energy agency) has an available shapefile of the status of dams in the country (in operation, proposed, at inventory phase, etc). The FHReD dataset also provides proposed dams worldwide, which includes many in SA (http://globaldamwatch.org/fhred/).

*Response:* Thank you for these relevant suggestions. We have improved section 6 'Summary' in order to include information regarding future perspectives for extending our database. First, we discuss information about future dams in South America. Additionally, we have included a Supplementary Table (Supplementary Table 1), which contains information about 245 future projects in South America, 61 under construction as of 2020 and 184 projects planned to be developed in the future. Supplementary Table 1 details future dams in South America identified by country, name and implementation phase.

Also, we present a discussion about additional attributes which could be included in future versions of our database, e.g. outflows of dams (discharge time series) or energy generation data from hydroelectric dams (energy generation time series):

*References to lines with the suffix 'OM' refer to the original manuscript and the refence to lines with the suffix 'RM' refer to the revised manuscript.*

**Line 284OM / 360RM:**

*'One of the main goals of this endeavour is to foster the research of water resources in South America. To achieve this objective, we consider that we must make the necessary efforts to keep our database relevant to the international hydrology community.*

*For this, we believe it will be necessary to keep our database updated, and also, include additional information regarding hydrology and water resources management in future versions of our database. Future dams are one of the topics we need to observe to maintain our database updated. In recent years, several South American countries have made public their intention to develop new dam projects, mainly for hydroelectric generation (Anderson et al., 2018; Moran et al., 2018; Zhang et al., 2018). We have identified 245 future projects in South America, 61 under construction for 2020 and 184 projects planned to be developed in the future. Supplementary Table 1 details future dams in South America identified by country, name and implementation phase.*

*Monitoring the development of future dams in South America is necessary due to the relevance of these projects on the local and regional scales. It is not likely that all projects listed in Supplementary Table 1 will be carried out due to different economic, social or political factors (Anderson et al., 2018). However, the likely ecological or social impacts that these projects may cause (Doria et al., 2018; Lees et al., 2016; Winemiller et al., 2016) highlight the necessity for the international hydrological community to be conscious of the status of these projects.*

*Similarly, we consider that future versions of our database may be extended with additional attributes. For example, information such as outflows of dams (discharge time series) or energy generation data from hydroelectric dams (energy generation time series), could also be included in the future. However, to date, including this type of information on a continental scale represents a significantly great effort due to the lack of readily available information on water resources in most countries of the region. There are countries, like Brazil, which make public their relevant information about water resources and energy generation through their official agencies, e.g. the National Agency of Water ANA (https://www.ana.gov.br/sar/sin, last access: 9 Nov 2020), and the National Electric Energy Agency ANEEL (https://www.aneel.gov.br/siga, last access: 9 Nov 2020). Then again, other countries of the region keep this information restricted or outdated, which makes it difficult to complete these attributes for the entire database.*

*Finally, the data presented in this database is largely based on open-access information available to date, therefore, the valuable support of both public institutions and the international hydrology community will be necessary for extending future versions of our database. This will allow us to keep our database relevant, which in turn will support the development of future research initiatives on water resources in the region.'*

3. The authors could consider presenting an updated map of the degree of regulation index (DoR; basically the total storage of upstream reservoirs divided by the average discharge at a given river reach) which is a simple one yet powerful to understand reservoir regulation at large drainage networks. This is easy to do, since the authors already have the Hydrosheds ID for each dam location. This would be a kind of updating for SA of the free-flowing rivers map published recently (Grill et al 2019 Nature).

*Response:* Thank you for this valuable suggestion. We have determined the degree of regulation for the dams in our database and included the results in our manuscript in sections: 'Abstract', '1 Introduction', '2.2.10 Degree of regulation', section '3.2 hydrological information', section '4 Data limitations' and also in figure 2f, figure 5 and figure 6.

*Line 17OM / 17RM:*

[revised manuscript text omitted]

4. The interpretation of the hydrological data and the outcomes of the dataset in the Results section is too simplistic. For example, in the section 3.1 Dams and Reservoirs there is only a comparison with GRanD and AQUASTAST databases. However, given the large amount of data available, more interesting figures as histograms with number of dams implemented per year and per country should be included. Regional analyses could also be performed, e.g., higher dams are mainly located in which countries, in which type of environment? Although I recognize that this is mainly a paper describing the dataset itself, some additional analyses could be included and would certainly improve the overall quality of the manuscript. In section 3.2 Hydrological Information, the authors focus on describing extreme values of PET, Precipitation, temperature and other variables at individual sites (e.g., 'The highest potential evapotranspiration record is documented for the catchment of the "Pilões" dam in Brazil with 1,713.32 millimetres per year'). However, for a continental scale dataset as this one, I think that regional analyses would be much more interesting, e.g., how many dams are located in regions with high aridity index (PET/P)? Similarly, in section '3.3 Additional Information', there is only a simple phrase on how Yaciretá dam is associated to the highest upstream population and equipped areas of irrigation. A more thorough analysis describing the distribution of dams at different levels of population pressure across the continent could be included.

The authors could consider analyzing upstream population divided by the dam drainage area, this would put some weight on the large upstream population for dams located in downstream reaches as Yaciretá dam in the Paraná river.

*Response:* Thank you for this suggestion we have made several improvements in our analysis. Besides the 'Degree of Regulation Index' explained in the previous section, we have included 2 additional indicators for our assessment: 'Aridity Index' and 'Residence Time'. We mention these indicators in sections: 'Abstract', '1 Introduction', '2.2.8 Aridity Index' and '2.2.9 Residence Time', '3.2 Hydrological Information', and '4 Data limitations and uncertainties'. We also have improved the entire section '3 Results'. We believe

these indicators and further assessment will allow us to clarify our results and improve the overall outcome of our manuscript.

We have also included several additional figures: 2a, 2b which depict an analysis about dam information (number, storage volume, country, year). Figure 2c assesses population and catchment data per each dam and country. Figure 2d evaluates dams per aridity index and per country and figure 2e assesses upstream runoff and residence time. In addition, figure 4 was updated to include the aridity index and the residence time per dam catchment.

**Line 17OM / 17 RM:**

[revised manuscript text omitted]

**Figure 2a (Figure 2 is a composition of figures 2a, 2b, 2c, 2d, 2e and 2f)**

[Figure]

*Figure a) Cumulative number of dams per decade and per storage volume*

**Figure 2b:**

[Figure]

*Figure b) Cumulative number of dams per decade per country*

*Figure 2c:*

[Figure]

*Figure c) Upstream population per catchment area and per country*

*Figure 2d:*

[Figure]

*Figure d) Number of dams per aridity index type and per country*

[Figure]

*Figure e) Annual catchment runoff per residence time and per country*

[Figure]

*Figure 4. a) Population, b) Equipped Area for Irrigation, c) Aridity Index, d) Residence Time per dam catchment.*

5. Why is 'equipped areas of irrigation' considered an 'additional information'? For me it is a hydrological information.

*Response:* Thank you for this suggestion. We have considered your suggestion appropriate and made an improvement on this entire section. We have removed the 'Additional information' in section 2 by combining the section '2.2 Data Sources' and the section '2.3 Data processing', into a new section '2.2 Data sources and assessment methods'. The relevant information in section 2.3 has been included in each of the groups in section 2.2. Finally, we have revised each of the groups in the new section 2.2 to include only the appropriate content.

*Line 91OM / 107RM:*

[revised manuscript text omitted]

6. More information on the data used (section 2.2) should be provided. For example, some information is missing, as the unit of catchment irrigation area (this is only presented in figure 3, and it not presented in the main manuscript or in the provided data in Zenodo).

*Response:* Thank you for this suggestion. Table 4 describes the list of variables processed for dams and reservoirs in our database, this table includes information about units and other relevant information. Also, file 4. Dataset Attribute Description in the Zenodo repository, includes detailed description for all attributes in the database. Nevertheless, we have made improvements in section 2.2 to include unit's information to each subsection.

*Line 441OM / 595RM:*

*Table 1: List of hydrological and additional information processed in this study*

| VARIABLE | UNIT | DESCRIPTION |
|---|---|---|
| CATCHMENT AREA | $km^2$ | Calculated catchment area per dam expressed in square kilometres |
| NEAR SURFACE TEM-PERATURE | °C | Calculated monthly average near surface temperature value derived from the Climatic Research Unit (CRU TS 4.03) time-series dataset per each dam catchment expressed in degrees Celsius |
| PRECIPITATION | mm/month | Calculated monthly average precipitation value derived from the Climatic Research Unit (CRU TS 4.03) time-series dataset per each dam catchment expressed in millimetres per month |
| POTENTIAL EVAPO-TRANSPIRATION | mm/day | Calculated monthly average potential evapotranspiration value calculated using the Pen-Monteith method derived from the Climatic Research Unit (CRU TS 4.03) time-series dataset per each dam catchment expressed in millimetres per day |
| GRDC | mm/month | Calculated monthly average monthly runoff derived from the University of New Hampshire Global Runoff Data Centre (GRDC) composite runoff field per each dam catchment expressed in millimetres per month |
| POPULATION | people | Calculated population data from the Global Rural-urban Mapping Project (GRUMP) per dam catchment |
| IRRIGATION | ha | Calculated irrigation area from the Global Map of Irrigated Area dataset per dam catchment expressed in hectares. |

*Line 145OM / 216RM:*

*'The units of EIA are expressed in square hectares (ha).'*

7. The authors use the catchments of each dam to estimate some properties (upstream population, etc). The catchment polygons are presented in Figures 2 and 3. I think a shapefile with the polygons should also be provided in the Zenodo dataset, what is very useful for users to extract other interesting information, and it would be in the context of other initiatives of hydrological datasets as CAMELS-Chile (Alvarez-Garreton et al 2018 HESS) and CAMELS-Brazil (Chagas et al 2020 ESSD).

*Response:* Thank you for this suggestion. In order to assist potential users of our database we uploaded a new version of the DDSA database, including the new attributes that have been processed: aridity index, accumulated upstream reservoir capacity, average discharge volume per year, and degree of regulation. Also, we included a shapefile with the catchment polygons of each dam.

Finally, some text clarifications are still required in some parts. Some paragraphs are also too long and must be reduced or splitted. I provide some minor suggestions below.

*Minor suggestions:*

8. Line 9 Split into two sentences: 'In general, its relevance relies on facilitating the management of water resources for anthropogenic purposes. However, dams could also generate many potential adverse impacts related to safety, ecology orbiodiversity.'

*Response:* Thank you for this suggestion. We have split the initial sentence into two sentences.

*Line 9OM / 9RM:*

*'...In general, its relevance relies on facilitating the management of water resources for anthropogenic purposes. However, dams could also generate many potential adverse impacts related to safety, ecology or biodiversity. ...'*

9. L.18 'dams' catchments'

*Response:* Thank you for this suggestion. We have corrected the writing in this phrase.

*Line 18OM / 18RM:*

*'...dams' catchments ...:'*

*Line 69RM:*

*'...dams' catchments...,'*

*Line 162RM:*

*'...the dams' catchments.'*

*Line 173RM:*

*'...the dams' catchments...'*

*Line 175RM:*

*'...the dams' catchments.'*

*Line 195OM / 191RM:*

*'...the dams' catchments...'*

*Line 203OM / 197RM:*

*'...the dams' catchments...'*

*Line 215OM / 208RM:*

*'...the dams' catchments...'*

**10.** L.23 'contribute to the development...'

*Response:* Thank you for this suggestion. We have corrected the writing of this phrase.

*Line 23OM / 23RM:*

*'... to contribute to the development...'*

**11.** L.33 'assess'

*Response:* Thank you for this suggestion. We have corrected the writing of this word.

*Line 33OM / 37RM:*

*'... that assess or...'*

**12.** L.49 'La Plata' instead of 'El Plata'

*Response:* Thank you for this suggestion. We have corrected the writing of this word.

*Line 49OM / 53RM:*

*'...La Plata...'*

**13.** L.52 'which reports' instead of 'and reports'

*Response:* Thank you for this suggestion. We have corrected the writing of this phrase.

*Line 52OM / 56RM:*

*'... which reports...'*

14. L.54'...America it only reports...'

*Response:* Thank you for this suggestion. We have corrected the writing of this phrase.

   *Line 54OM / 58RM:*

   *'...America it only reports...'*

15. L.72 check:'5,283,000'

*Response:* Thank you for this suggestion. We have corrected the writing of numbers.

   *Line 72OM / 77RM:*

   *'...5,283,000 ...'*

16. L.74 'Paraná' with acute accent

*Response:* Thank you for this suggestion. We have corrected the writhing of this word.

   *Line 49OM / 53RM:*

   *'... Paraná ...'*

   *Line 74OM / 83RM:*

   *'...Paraná ...'*

17. L.81'...the continent, there exist humid...'

*Response:* Thank you for this suggestion. We have corrected the writing of this phrase.

   *Line 81OM / 83RM:*

   *'...the continent, there exist humid...'*

18. L.84'found' instead of 'find'

*Response:* Thank you for this suggestion. We have corrected the writing of this word.

   *Line 84OM / 87RM:*

   *'...are found ...'*

19. L.85'...Chile, which are blocked due to the Andes mountains, which causes low precipitation...'

*Response:* Thank you for this suggestion. We have corrected the writing of this phrase.

   *Line 85OM / 88RM:*

   *'... Chile, which are blocked due to the Andes mountains, which causes low precipitation...'*

20. L.88'and it is located'

*Response:* Thank you for this suggestion. We have corrected the writing this phrase.

   *Line 88OM / 91RM:*

   *'...and it is located in ...'*

21. L.89'for example the "El Niño",...'

   *Response:* Thank you for this suggestion. We have corrected the writing of this phase.

   *Line 89OM / 94RM:*

   *'..., the "El Niño Southern Oscillation" (ENSO)...'*

   *Line 98RM:*

   *'...the "El Niño" ...'*

   *Line 99RM:*

   *'...the "El Niño" ...'*

22. L.89-90 this whole phrase is confusing, please rephrase. Besides, it is too simplistic to state that ENSO 'increases precipitation at the northwest area' since it affects in very different ways different regions of South America. Please improve this description here.

   *Response:* Thank you for this suggestion. We have improved these paragraphs in order to improve the description of climate events in South America.

   *Line 89OM / 93RM:*

   *'Climate diversity in South America is also due to the occurrence of several interannual and inter-decadal large-scale climate events. For example, the "El Niño Southern Oscillation" (ENSO) which is a Pacific Ocean sea-surface temperature (SST) event that fluctuates from warm ("El Niño") and cold ("La Niña") phases, and occurs in periods of between two to seven years. The ENSO causes disruptions of precipitation and temperature in the continent and is often considered as the major source of interannual climate variability in most of South America.*

   *In general, the "El Niño" causes low precipitation over tropical South America, high precipitation over the south east of the region and high temperatures over tropical and subtropical areas. Also, the "El Niño" is often associated to regionally diverse events like droughts in the Amazon rainforest and the north-east of South America, but also to flooding events in the tropical west coast and the south-east of the continent (Cai et al., 2020; Hao et al., 2020). On the other hand, "La Niña" generally causes the opposite precipitation and temperature events for the same areas (Garreaud et al., 2009).*

   *Other regional climate events in South America like the sea-surface temperature (SST) anomalies in the tropical Atlantic (Garreaud et al., 2009; Jiménez-Muñoz et al., 2016), the Pacific Decadal Oscillation (PDO) (Nathan and Steven, 2002), or the Antarctic Oscillation (AAO) and the North Atlantic Oscillation (NAO) (Garreaud et al., 2009) also play an important role in the variability of South America climate.'*

23. L.96 'The GRanD' - 'The' should be in lowercase.

   *Response:* Thank you for this suggestion. We have corrected the writing of this word.

   *Line 96OM / 113RM:*

   *' the GRaND database...'*

24. L.112 a reference for HydroSHEDS should be included (Lehner et al), not only the dataset website

*Response:* Thank you for this suggestion. We have included the corresponding reference to the Hy-droSHEDS dataset.

> ### *Line 112OM / 155RM:*
>
> *'...the HydroSHEDS (Hydrological data and maps based on SHuttle Elevation Derivatives at multiple Scales) (Lehner et al., 2008) dataset ...'*

25. L.119 a reference for CRU should be included (New et al)

*Response:* Thank you for this suggestion. We have included the corresponding reference to the CRU da-taset. We should point out that we used the reference (Harris et al., 2020) because this is the reference that that authors mention in the dataset website for their latest versions. (https://crudata.uea.ac.uk/cru/data/hrg/#current).

> ### *Line 119OM / 169RM:*
>
> *' This data was derived from the Climatic Research Unit (CRU) time-series dataset (Harris et al., 2020),'*

26. L.127 a reference for GRDC should be included

*Response:* Thank you for this suggestion. We have included the corresponding reference to the GRDC dataset.

> ### *Line 127OM / 185RM:*
>
> *'We used the University of New Hampshire and Global Runoff Data Centre (UNH/GRDC) Composite Runoff field v1.0 (Fekete et al., 2002),...'*

27. L.135 a reference for GRUMP should be included

*Response:* Thank you for this suggestion. We have included the corresponding reference to the GRUMP dataset.

> ### *Line 135OM / 198RM:*
>
> *'... the Global Rural-urban Mapping Project (GRUMP) (Center for International Earth Science Information Network CIESIN et al., 2011).'*

28. L.135 'for each of dams' catchments...'

*Response:* Thank you for this suggestion. We have corrected this phrase.

> ### *Line 135OM / 197RM:*
>
> *'...the dams' catchments...'*

29. L.141 'catchment were extracted'

*Response:* Thank you for this suggestion. We have corrected this phrase.

> ### *Line 141OM / 208RM:*
>
> *'... for each of the dams' catchments were extracted ...'*

**30.** L.180 'reservoirs' catchments' instead of 'reservoir's catchments': please check this throughout the whole text. The 'catchments' refer to all 'reservoirs', and not just to one reservoir and stated in the current form 'reservoir's catchments'

*Response:* Thank you for this suggestion. We have checked throughout the manuscript to improve the writing of this phrase. On the other hand, as mentioned above section title '2.3.2 Hydrological information of the reservoir's catchments' in line 180 has been removed due to an improvement on this section.

**31.** L.195 'performed' instead of 'calculated'

*Response:* Thank you for this suggestion. We have improved the writing in this phrase.

> *Line 195OM / 173RM:*
>
> *'Finally, we computed the long-term mean monthly values for…'*

**32.** L.195 which statistical analysis was performed? or was it just a long-term average for each month?

*Response:* Thank you for this suggestion. We have improved the writing in this phrase.

> *Line 195OM / 173RM:*
>
> *'Finally, we computed the long-term mean monthly values for…'*

**33.** L.225'...of the data was concluded, ...'

*Response:* Thank you for this suggestion. We have improved the writing in this phrase.

> *Line 225OM / 258RM:*
>
> *'...of the data was concluded …'*

**34.** L.225-226 phrase too long, please reduce it or split into two phrases.

*Response:* Thank you for this suggestion. We have improved this phrase and divided it into two phrases.

> *Line 225OM / 258RM:*
>
> *'Once the review, refinement and processing of the data was concluded, a total of 1,010 dam entries were accepted for our database (Figure 1). This represents a noticeable progress in the identification and geolocation of dams in the region and thus, enables the opportunity for new research that allows a more precise understanding of the water resources systems in the region.'*

**35.** L.228 'GRanD' instead of 'GrAND' - please check throughout the text

*Response:* Thank you for this suggestion. We have checked throughout the manuscript to improve the writing of this term.

> *Line 55OM / 59RM:*
>
> *'...GRaND…'*
>
> *Line 97OM / 113RM:*
>
> *'...GRaND…'*
>
> *Line 158OM / 125RM:*

*'...GRaND...'*

***Line 164OM / 130RM:***

*'...GRaND...'*

***Line 228OM / 261RM:***

*'...GRaND...'*

***Line 263OM / 267RM:***

*'...GRaND...'*

36. L.240 '14,855,192'

    *Response:* Thank you for this suggestion. We have corrected the writing of numbers.

    ***Line 240OM / 280RM:***

    *'...14,855,192...'*

37. L.240 please split phrase in two:'...kilometres. The largest catchments...'

    *Response:* Thank you for this suggestion. We have improved this phrase and divided it into two phrases.

    ***Line 240OM / 280RM:***

    *'... square kilometres. The largest catchment ...'*

38. L.242 'Our results highlight the great influence and importance of the Amazon rainforest in the continent since most of the highest records': I do not understand the relevance of this phrase for the context of a database of dams.

    *Response:* Thank you for this suggestion. We have removed this phrase and improved the understanding in this section.

    ***Line 242OM / 280RM:***

    *'...to the "Jirau" dam in Brazil with an estimated area of 962,732 km2. Table 4 describes the variables processed for the hydrological information included in this database.*

    *Figure 3 presents the annual values for NST, P, PET and runoff estimated for each dam catchment. Both in the case of NST and P, higher values would seem to be mostly located near the equator, while PET higher values are more noticeable in the northeast of Brazil. In the case of runoff, values are scattered and there is no evident predominance, except for higher values localized in the southeast of Brazil.'*

39. L.248 this runoff value of 2961 mm/year for Billings catchment is certainly a model error, since it does not rain that much in this catchment to have this runoff. The high precipitation rates occur more in the mountains close to São Paulo. You can check it in the Brazilian precipitation maps by the Brazilian Geological Survey (CPRM): <http://www.cprm.gov.br/publique/Hidrologia/Mapas-e-Publicacoes/Atlas-Pluvio-metrico-do-Brasil-1351.html>. The runoff model uncertainty should be discussed here. I honestly

*Response:* Thank you for this valuable suggestion. We have taken note of your observation and have verified our runoff model. Our model derives the runoff value from the cells of the UNH/GRDC dataset that are within the catchment area of each dam.

In the case of the Billings catchment, two particular situations are observed: The first is that the catchment area of this dam is smaller than the individual cell area of the UNH/GRDC dataset (0.5x0.5 decimal degrees), which prevents our model to sample enough cells to estimate a more accurate result than those which could be derived from a local study. The second is that the Billings dam catchment area is located between two cells of the UNH/GRDC dataset: the first cell is in the area of São Paulo, where the dam is located, and the other cell is located in the mountainous area near São Paulo, which is an area with high precipitation values. This second cell is where a significant part of the catchment area is located and thus, the source from the majority of the runoff value for this catchment.

After reassessing our model, we consider that the uncertainty in the runoff value computed for the Billings dam has a low probability of occurrence for other dams in our database.

However, having evidenced this situation, we have considered it necessary to include an improvement to section '3.2 Hydrological information' and section '4 Data limitation and uncertainties', mentioning the potential limitations of our hydrological inputs.

**Line 248OM / 280RM:**

*'...to the "Jirau" dam in Brazil with an estimated area of 962,732 km2. Table 4 describes the variables processed for the hydrological information included in this database.*

*Figure 3 presents the annual values for NST, P, PET and runoff estimated for each dam catchment. Both in the case of NST and P, higher values would seem to be mostly located near the equator, while PET higher values are more noticeable in the northeast of Brazil. In the case of runoff, values are scattered and there is no evident predominance, except for higher values localized in the southeast of Brazil.'*

**Line 259OM / 316RM:**

*'**4 Data limitations and uncertainties***

*The information provided in this database cannot be considered error free since it has been prepared using the information available at the time of its elaboration. It should also be noted that although our database was created independently, through an individual investigation and based primarily on reports and documents available from each of the countries in the region, the database may include attributes of dams that are also reported by other existing dam databases such as ICOLD, AQUASTAT and GRaND.*

*Hydrological inputs provided in this database also need careful interpretation to avoid misleading interpretations. First, the resolution of the hydrological datasets used in the DDSA database could affect the accuracy of results for small catchments. Although all the datasets considered in this database have been largely validated for large-scale or regional assessments models (Gonzàlez-Zeas et al., 2012; Lv et al., 2018), we suggest caution if the intention is to use these results in catchments with an area smaller than the cell size of each dataset.*

*Our results regarding aridity index, residence time and degree of regulation also need to be interpreted with caution. First, our results are intended to assess the dams' catchments and therefore, should be used carefully if intended for other type of assessment. Also, in the case of the DOR index, there are many important inputs in our assessment that may have not been considered due to the*

*absence of information. For example, we are not considering information about local water use, specific stream characteristics or relevant and updated urban information. Also, our DOR assessment does not consider unidentified small reservoirs, which could alter the final results. Furthermore, the impacts of river regulation also depend on a wide range of factors, e.g. local or international policies, which have not been considered either. Altogether, we consider that despite the aforementioned uncertainty factors, our results give a consistent first approximation of these indices at a regional scale.*

*Finally, in order to assess the robustness of our DOR assessment, we conducted a sensitivity analysis by comparing our findings with the results determined by (Grill et al., 2019) in their manuscript 'Mapping the world's free-flowing rivers' (DoR_FFR). Figure 6 compares 409 stream matches from both studies and determines a strong correlation (r=0.702) between our results and the DoR_FFR manuscript. The correlation results are more evident on large and very large rivers.'*

**_40._** L.450 I do not understand why Figure 3 has figures e) and f), and not a) and b), since it is a figure by itself, and not a continuation of Fig 2.

***Response:*** Thank you for this suggestion. We have updated figure 4 to correct the numbering of the maps on the Figure and also include data from aridity index and residence time.

[revised manuscript text omitted]

---

## Author Comment (AC2) · 20 Nov 2020

**Response to Anonymous Referee #2**

_1._ This manuscript entitled "Dataset of Georeferenced Dams in South America (DDSA)" clearly explains the importance of a database of georeferenced dams in South America, along with other variables that describe each local dam scenario. Thus, the purpose of the proposed work is clear and important in order to establish a base line of information for other studies related to dams and reservoirs. I think the idea and work are in concordance with the journal's and I encourage the authors to modify the text and also include guidelines for future database modifications (i.e. updates). I recommend this article for publication upon the following major revisions:

_**Response:**_ The authors take the opportunity to acknowledge the valuable comments provided by the anonymous referee, as well as the time that has been committed to provide this valuable feedback. All suggestions made have been considered and addressed in a reasoned manner. Revisions have been made to the manuscript and are described below.

_2._ The relevance of the proposed database for future studies will be directly linked with latest update. Thus, a mechanism for complementing the information should be provided. What should other researchers do in order to update the data on a given region?

_**Response:**_ Thank you for this suggestion. We have improved section 5 in order to encourage interested researchers to make whatever contributions they deem necessary to keep our database up to date. Likewise, we have described how to access the database through the free access repository ZENODO and also, we have included in the repository access link, the contact information of the authors, in order to receive the contributions from the research community.

_References to lines with the suffix 'OM' refer to the original manuscript and the refence to lines with the suffix 'RM' refer to the revised manuscript._

> _**Line 265OM / 339RM:**_
>
> _**'5 Data availability**_
>
> _The Database of Georeferenced Dams of South America (DDSA) is a joint effort of researchers from the Department of Civil Engineering: Hydraulics, Energy and Environment of the Universidad Politécnica de Madrid and the Civil Engineering Career of the Universidad Técnica de Ambato. The DDSA database is available for both researchers and the general public through the ZENODO open access repository https://doi.org/10.5281/zenodo.3885280 (Paredes-Beltran et al., 2020), where we have detailed the contact information of the authors, in order to receive any valuable contribution which could allow us to improve our database.'_

_3._ The Data description section is divided in three subsections: 2.1. Study Area. 2.2. Data Sources, and 2.3. Data Processing. Subsections 2.2 and 2.3 repeat a significant portion of the information. I would suggest to the authors to fusion these two into a single subsection "Data sources and assessments methods" maintaining the 7 groups originally indicated in subsection 2.2 and including within each of them the methods used for the respective data assessment.

_**Response:**_ Thank you for this suggestion. We have reviewed your suggestion and found it appropriate. We have combined the section '2.2 Data Sources' and the section '2.3 Data processing' into a new section '2.2 Data sources and assessment methods'. The relevant information in section 2.3 has been included in each

of the groups in section 2.2. Finally, we have revised each of the groups in the new section 2.2 to include only the appropriate content.

*Line 91OM / 107RM:*

[revised manuscript text omitted]

**4.** Line 25. Mentions that dams are, in many cases, controversial due to "some associated negative impacts…" and the following sentence indicates that "these structures cause major impacts and changes wherever these are implemented"- this is ambiguous. Please refer to the impacts complementing line 40 in which a list of them is considered and complement it with other topics such as, morphology, water quality and habitat. Please classify them as acute and chronic impacts (time related impacts) since it will add value to the continuous monitoring efforts.

*Response:* Thank you for this suggestion. We have improved the two paragraphs mentioned in order to avoid ambiguity. Also, we have supplemented these sections with information on the impacts caused by dams and reservoirs.

**Line 25OM / 25RM:**

*'1 Introduction*

*Dams and their reservoirs provide continuous water supply for different anthropogenic necessities such as electricity generation, water supply, irrigation, flood control, livestock feed or recreation. This becomes crucial in areas where water resources are scarce either by seasonality or due to the increasing effects of climate change. However, in many cases dams and their reservoirs are controversial because they can cause acute and chronic impacts in the environment and also in the nearby human settlements. These impacts are generally well known and include the modification of aquatic and terrestrial ecosystems, reduction of biodiversity, changes in the morphology of river systems, degradation of water quality and characteristics, alterations in sediments and nutrients discharge, changes in seasonal hydrological regimes, the migration of human settlements or changes in land-use patterns (Barbarossa et al., 2020; Bednarek, 2001; Nilsson et al., 2005; Pekel et al., 2016; Stoate et al., 2009).*

*Due to the obvious importance of dams and their reservoirs, continuous monitoring and resources needs to be dedicated on these structures. The importance of dams and reservoirs also makes them relevant for research. For example, there are studies that assess or propose improvements on construction methods for dams (Ladd, 1992; Noorzaei et al., 2006; Xu et al., 2012), examine improvements on monitoring the structural health or safety of the dam (Gabriel-Martin et al., 2017; Li et al., 2004; Sjödahl et al., 2008) or evaluate their behaviour during seismic or failure events (Alonso et al., 2005; Zabala and Alonso, 2011). Reservoirs associated with dams are also relevant, for instance, by examining the effects, impacts and management alternatives of sediments fluxes (Dai and*

*Liu, 2013; Kondolf et al., 2014). Usually, these studies require knowing a minimum set of characteristics of the dam, including their location and in most of the cases, need to be included into hydrological models.'*

5. Line 65. Please explain which "fields" the authors are referring to.

   *Response:* Thank you for this suggestion. We have improved this section in order to better describe the research fields where we think our database could be used.

   ### *Line 65OM / 68RM:*

   *'This database has been developed to provide researchers additional information on dams, reservoirs and dams' catchments in South America, with the expectation to further promote research on dams, hydrology, water resources, ecology environmental science, geography or sociology either on a local, regional or global scale.'*

6. Line 70 and 75 mention the Amazon, Parana-Rio de la Plata, and the Orinoco rivers as the largest systems in the region. Please revise to consider mentioning these fluvial systems only once when describing the study area.

   *Response:* Thank you for this suggestion. We have improved these paragraphs in order to avoid repeated sentences.

   ### *Line 70OM / 74RM:*

   *'2.1 Study Area*

   *The study area is the continent of South America and includes Argentina, Bolivia, Brazil, Chile, Colombia, Ecuador, Guyana, French Guiana, Paraguay, Peru, Suriname, Uruguay and Venezuela. A total of 1,010 catchments were considered which drain an area of approximately 5,283,000 km2 and discharge their waters to both the Pacific Ocean and the Atlantic Ocean. Within each of this catchments, necessary observations were made to accurately locate dams with their respective reservoirs.*

   *The study area is diverse and full of contrasts due to its unique geography; for example, the Andes mountains are a continuously seismic region that covers the entire west coast of the continent, the Amazon rainforest in the central part of the continent, large semiarid plains in the southeast and also the Atacama desert, which is a region of extreme aridity in the southwest. In the Andes we have the presence of large glaciers that mostly drain east to form several rivers, including some of the largest in the world such as the Amazon, the Paraná - Rio de la Plata and the Orinoco river. On the east coast of the continent, there exist humid mountain formations that extend from Venezuela to northern Brazil.'*

7. Line 85 refers to "El Niño" however there is no mention to "La Niña" that also brings changes to the precipitation patterns and it should be considered.

   *Response:* Thank you for this suggestion. We have improved these paragraphs in order to improve the description of climate events in South America. We have included a description of the ENSO, as well as we have made mention of other relevant climatological events in the region.

   ### *Line 85OM / 93RM:*

*'Climate diversity in South America is also due to the occurrence of several interannual and inter-decadal large-scale climate events. For example, the "El Niño Southern Oscillation" (ENSO) which is a Pacific Ocean sea-surface temperature (SST) event that fluctuates from warm ("El Niño") and cold ("La Niña") phases, and occurs in periods of between two to seven years. The ENSO causes disruptions of precipitation and temperature in the continent and is often considered as the major source of interannual climate variability in most of South America.*

*In general, the "El Niño" causes low precipitation over tropical South America, high precipitation over the south east of the region and high temperatures over tropical and subtropical areas. Also, the "El Niño" is often associated to regionally diverse events like droughts in the Amazon rainforest and the north-east of South America, but also to flooding events in the tropical west coast and the south-east of the continent (Cai et al., 2020; Hao et al., 2020). On the other hand, "La Niña" generally causes the opposite precipitation and temperature events for the same areas (Garreaud et al., 2009).*

*Other regional climate events in South America like the sea-surface temperature (SST) anomalies in the tropical Atlantic (Garreaud et al., 2009; Jiménez-Muñoz et al., 2016), the Pacific Decadal Oscillation (PDO) (Nathan and Steven, 2002), or the Antarctic Oscillation (AAO) and the North Atlantic Oscillation (NAO) (Garreaud et al., 2009) also play an important role in the variability of South America climate.'*

8. Line 100 cites Table 1 regarding the government's available information. Please include the respective links to the information. This is important in order to replicate efforts in the future.

*Response:* Thank you for this suggestion. We have improved the description of Table 1 in order to include an additional column with the reference's information at the end of the table. We have also included the link to each source in the references section. This was due to the fact that we considered that several of the links were too long, so we preferred to mention the links in the references section to improve the visualization of the table.

Table 1: Available public data records of dams per country

| COUNTRY | AVAILABLE PUBLIC INFORMATION | NUMBER OF ENTRIES | GEOREFERENCED INFORMATION | REFERENCE* |
|---|---|---|---|---|
| ARGENTINA | Inventario de Presas y Centrales Hidroeléctricas de la República Argentina | 31 | No | (Subsecretaría de Recursos Hídricos, 2010) |
| BOLIVIA | Inventario Nacional de Presas Bolivia | 287 | Yes | (Programa de Desarrollo Agropecuario Sustentable (PROAGRO), 2010) |
| BRAZIL | Mapa Interativo das Barragens Cadastradas | 18,880 | Yes | (Agencia Nacional de Aguas, 2019) |
| CHILE | Directorio de Presas | 107 | No | (Comite Nacional Chileno de Grandes Presas, 2019) |
| COLOMBIA | ISAGEN | - | No | (ISAGEN, 2020) |
| ECUADOR | Various web pages | - | No | (ElecAustro, 2020; Hidroabanico, 2019; Instituto Nacional de Pesca, 2020) |
| FRENCH GUIANA | Not available | - | - | |
| GUYANA | Not available | - | - | |
| PARAGUAY | Not available | - | - | |
| PERÚ | Inventario de Presas en el Perú | 743 | Yes | (Autoridad Nacional del Agua, 2016) |
| SURINAME | Not available | - | - | |
| URUGUAY | Not available | - | - | |
| VENEZUELA | Other | - | No | (Instituto Nacional de Estadistica, 2020) |

* The data records of each country website links are detailed in the reference section

*Line 257OM / 398RM:*

*Agencia Nacional de Aguas (2019) Mapa Interativo das Barragens Cadastradas no Sistema, Sistema Nacional de Informações sobre Segurança de Barragens SNISB. Available at: http://www.snisb.gov.br/portal/snisb/mapas-tematicos-e-relatorios/mapa-interativo-das-barragens-cadastradas (Accessed: 11 November 2020).*

*Line 302OM / 407RM:*

*Autoridad Nacional del Agua (2016) Inventario de Presas en el Peru. Lima. Available at: https://www.ana.gob.pe/etiquetas/inventario-de-presas.*

*Line 328OM / 442RM:*

*Comite Nacional Chileno de Grandes Presas (2019) Icold Chile - Directorio de Presas. Available at: http://www.icoldchile.cl/directorio/ (Accessed: 21 November 2019).*

*Line 334OM / 453RM:*

*ElecAustro (2020) Represa El Labrado. Available at: https://www.elecaustro.gob.ec/ (Accessed: 9 February 2020).*

*Line 361OM / 486RM:*

[revised manuscript text omitted]

---

## Author Comment (AC3) · 20 Nov 2020

Dear Referee:

Please find in the attached supplement of this commentary the following files: a)Supplementary Table 1 'Future Dams in South America', b) Figure 2a, c) Figure 2b, d) Figure 2c, e) Figure 2d, f) Figure 2e, g) Figure 2f, h) Figure 4 i) Figure 5 j) Figure 6.

The files mentioned above mentioned form part of our referee response.

Sincerely,

The authors.

[Figure]

Please also note the supplement to this comment:
https://essd.copernicus.org/preprints/essd-2020-188/essd-2020-188-AC3-supplement.zip

---

## Author Response (AR2)

**Dear Editor:**

We thank you for this opportunity to revise and improve our manuscript. We have considered all comments made by the Anonymous Referee and modified the manuscript with real thought as to make it more interesting and relevant to the readers of Earth System Science Data. This revised version includes significant improvement and addresses all comments expressed by the Anonymous Referee.

Please find below a detailed description of our response to the comments made by the Anonymous Referee. We gratefully acknowledge the helpful comments that have contributed to the improvement of our paper.

Sincerely,

The authors.

**Response to Anonymous Referee #1**

The authors have largely improved the manuscript and I support its publication in ESSD.

*Response:* The authors take the opportunity to acknowledge the valuable comments provided by the Anonymous Referee #1, as well as the time that has been committed to provide this valuable feedback. All suggestions made have been considered and addressed in a reasoned manner. Revisions have been made to the manuscript and are described below.

***Minor revisions:***

1. The DoR index should be used with active storage instead of total reservoir storage. However, if active storage data are not available, it is understandable to use the total reservoir storage values. Please just make it clear and discuss it briefly in the text.

*Response:* Thank you for this relevant suggestion. We have used the total reservoir storage values instead of the active storage data because this information is not available for many dams. We have improved section 4 'Data limitations and uncertainties' in order to briefly discuss this issue.

References to lines with the suffix 'RM' refer to the revised manuscript, the refence to lines with the suffix 'TCM' refer to the revised manuscript with the track changes option activated.

**Line 329RM / 329TCM:**

*'Also, in the case of the DOR index, there are many important inputs in our assessment which have not been considered and may alter the assessment results. For example, given the scale of this study, we are not considering information about local water use, specific stream characteristics or relevant and updated urban information. Also, two relevant inputs were not considered in our DOR assessment: unidentified small reservoirs, and the reservoir's active storage instead of total reservoir storage. These inputs should be considered in order to obtain more accurate results of the flow regulation but were not considered due to the absence of this information. Furthermore, the impacts of river regulation also depend on a wide range of factors, e.g. local or international water management policies, which have not been considered either. Altogether, we consider that despite the aforementioned uncertainty factors, our results give a consistent first approximation of these indices at a regional scale.'*

2. Line 290 correct "Yayreta" by "Yacyretá"

*Response:* Thank you for this suggestion. We have corrected this word and revised the accents in the manuscript.

**Line 290RM / 290TCM:**

*Yacyretá*

**Line 291RM / 291TCM:**

*Itaipú*

**Line 294RM / 294TCM:**

*Yacyretá*

3. About the '245 future projects in South America', Almeida et al (2019 Nature Communications, https://www.nature.com/articles/s41467-019-12179-5) identified 351 proposed dams for the Amazon basin. This may be worth mentioning here. I could not find the Supplementary Table 1 with the list of future projects in the attached files, but I encourage the authors to double-check this '245' value.

*Response:* Thank you for this relevant suggestion. We are sorry that you were not able to find the Supplementary Table 1 in the attached files. The Supplementary Table 1 was uploaded as a supplementary file in the authors response stage and was also attached in the Author Comment 3: 'Response_referee1_files' which was uploaded on November 20 on the Interactive Discussion forum. Likewise, this file is available in the Zenodo repository, which is also mentioned in the manuscript. *We are including the Supplementary List 1 in the Supplement section as a PDF file.*

We have updated Supplementary Table 1, which now includes 574 future projected dams in South America, 61 under construction for 2020 and 513 planned projects for the future. This updated list includes all future projects identified by Almeida et al. 2019.

**Line 369RM / 365TCM:**

*'... (Almeida et al., 2019; Anderson et al., 2018; Moran et al., 2018; Zhang et al., 2018). We have identified 574 future projects in South America, 61 under construction for 2020 and 513 projects planned to be developed in the future. Supplementary Table 1 details future dams in South America identified by country, name and implementation phase.'*

**REFERENCES:**

Almeida, R. M., Shi, Q., Gomes-Selman, J. M., Wu, X., Xue, Y., Angarita, H., Barros, N., Forsberg, B. R., García-Villacorta, R., Hamilton, S. K., Melack, J. M., Montoya, M., Perez, G., Sethi, S. A., Gomes, C. P. and Flecker, A. S.: Reducing greenhouse gas emissions of Amazon hydropower with strategic dam planning, Nat. Commun., 10(1), doi:10.1038/s41467-019-12179-5, 2019.

Anderson, E. P., Jenkins, C. N., Heilpern, S., Maldonado-Ocampo, J. A., Carvajal-Vallejos, F. M., Encalada, A. C., Rivadeneira, J. F., Hidalgo, M., Cañas, C. M., Ortega, H., Salcedo, N., Maldonado, M. and Tedesco, P. A.: Fragmentation of Andes-to-Amazon connectivity by hydropower dams, Sci. Adv., 4(1), 1–8, doi:10.1126/sciadv.aao1642, 2018.

Moran, E. F., Lopez, M. C., Moore, N., Müller, N. and Hyndman, D. W.: Sustainable hydropower in the 21st century, Proc. Natl. Acad. Sci. U. S. A., 115(47), 11891–11898, doi:10.1073/pnas.1809426115, 2018.

Zhang, X., Li, H. Y., Deng, Z. D., Ringler, C., Gao, Y., Hejazi, M. I. and Leung, L. R.: Impacts of climate change, policy and Water-Energy-Food nexus on hydropower development, Renew. Energy, 116, 827–834, doi:10.1016/j.renene.2017.10.030, 2018.